# Noether Embedding:
# Efficient Learning of Temporal Regularities

**Chi Gao**[†]     **Zidong Zhou**[†]     **Luping Shi**[*]
Center for Brain-Inspired Computing Research,
Optical Memory National Engineering Research Center,
Tsinghua University - China Electronics Technology HIK Group
Co. Joint Research Center for Brain-Inspired Computing,
IDG / McGovern Institute for Brain Research at Tsinghua University,
Department of Precision Instrument,
Tsinghua University, Beijing 100084, China.
{gaoc20, zzd21}@mails.tsinghua.edu.cn
lpshi@mail.tsinghua.edu.cn

## Abstract

Learning to detect and encode temporal regularities (TRs) in events is a prerequisite for human-like intelligence. These regularities should be formed from limited event samples and stored as easily retrievable representations. Existing event embeddings, however, cannot effectively decode TR validity with well-trained vectors, let alone satisfy the efficiency requirements. We develop Noether Embedding (NE) as the first efficient TR learner with event embeddings. Specifically, NE possesses the intrinsic time-translation symmetries of TRs indicated as conserved local energies in the embedding space. This structural bias reduces the calculation of each TR validity to embedding each event sample, enabling NE to achieve data-efficient TR formation insensitive to sample size and time-efficient TR retrieval in constant time complexity. To comprehensively evaluate the TR learning capability of embedding models, we define complementary tasks of TR detection and TR query, formulate their evaluation metrics, and assess embeddings on classic ICEWS14, ICEWS18, and GDELT datasets. Our experiments demonstrate that NE consistently achieves about double the F1 scores for detecting valid TRs compared to classic embeddings, and it provides over ten times higher confidence scores for querying TR intervals. Additionally, we showcase NE's potential applications in social event prediction, personal decision-making, and memory-constrained scenarios.

## 1 Introduction

Recall the last time you went to a restaurant but waited for half an hour after ordering dishes. You probably knew something was wrong and may have called the waitperson for help. This behavior is guided by the temporal regularity (TR) of 'order dishes –(about 10 minutes)–> have meals' stored in your brain as schemas (Ghosh & Gilboa, 2014). Such TRs play a significant role in enabling humans to exhibit flexible out-of-distribution and systematic generalization abilities (Goyal & Bengio, 2022), and are directly learned from experience through a statistical accumulation of common event structures (Pudhiyidath et al., 2020), as shown in Figure 1. Since there exist enormous potential TRs due to a large number of event types and time intervals, detecting valid TRs from all

---

[†]Equal contribution.
[*]Corresponding author.
The code is publicly available at: https://github.com/KevinGao7/Noether-Embedding.

potential ones is therefore necessary, serving as a prerequisite capability for humans to form more complex event schemas in the brain to support downstream cognitive functions (Schapiro et al., 2017; McClelland et al., 1995). To attain human-level abilities in event scenarios, it is crucial to possess two fundamental properties when learning TRs. Firstly, TRs should be formed from even a limited number of experiences, which humans achieve since childhood (Pudhiyidath et al., 2020). Secondly, TRs should be stored as representations that can be instantly retrieved given appropriate cues, which is a central feature of human memory (Chaudhuri & Fiete, 2016).

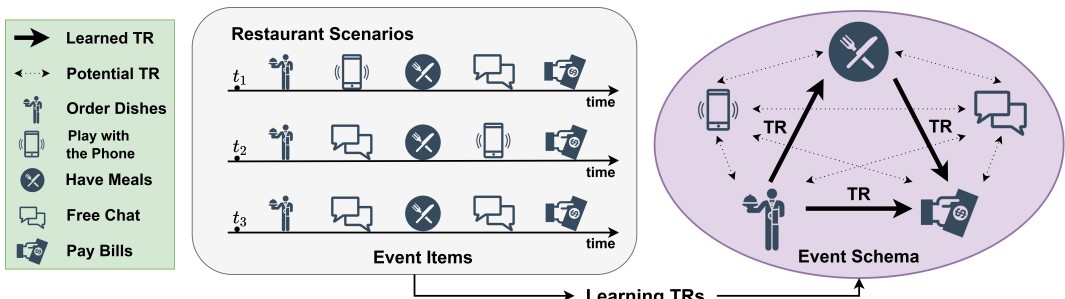

Figure 1: Illustration of TR learning. TRs indicate temporal associations invariant to time shifts, and are learned from event items through statistical accumulation.

It remains an open problem to jointly achieve the data-efficient formation and time-efficient retrieval of 1-1 TRs with event embeddings, although embedding models have achieved outstanding performance in various tasks such as event completion and event prediction (Cai et al., 2022; Zhao, 2021). Classic event embeddings can only encode patterns such as inversion and composition and decode the fitted event occurrences for better performance in completion tasks (Xu et al., 2020b; Wang et al., 2020; Messner et al., 2022). However, we aim to encode TRs by directly training the embeddings of each event sample and decode TRs by calculating well-trained embeddings without first decoding the fitted event occurrences. Our primary challenge in achieving such a counterintuitive function is to design the inductive bias that automatically integrates the event statistics of each potential TR over time.

Symmetry governs regularities in nature (Tanaka & Kunin, 2021), long before Noether proved the equivalence between symmetries and conservation laws in a physical system (Noether, 1918). Inspired by Noether's theorem, we develop Noether Embedding (NE) with the intrinsic time-translation symmetries of TRs indicated as conserved local energies in the embedding space. Calculating the event statistics of each potential TR is therefore converted to reducing the training loss of all event embeddings. This allows the direct revelation of TR validity by decoding the corresponding local energies through calculating well-trained embeddings after training convergence.

Contributions are twofold. Firstly, we develop NE which for the first time jointly achieves the data-efficient formation and time-efficient retrieval of TRs solely by embedding event samples. Secondly, we define complementary tasks of TR detection and TR query, formulate their evaluation metrics, and adopt classic datasets for evaluations, aiming at complete evaluations of embeddings' TR learning capabilities. Both our tasks and method generalize to arbitrary forms of structured events.

## 2  Problem Formalization

### 2.1  Definitions

**Temporal Regularity (TR)**   An event item $q$ can generally be represented by the basic symbolic form of $(ev, t)$, where $ev$ is the event type, and $t$ is the discrete occurrence time. Building on the interpretations from cognitive science literature (Ghosh & Gilboa, 2014; Pudhiyidath et al., 2020), we formally define TRs as temporal associations that remain invariant to time shifts:

$$(ev_b, t) \rightarrow (ev_h, t + \tau) \quad \forall t \in \mathbb{T}_a \tag{1}$$

$ev_b \neq ev_h, t, t + \tau$ respectively refer to the body and head event type and their occurrence time. $\mathbb{T}_a$ refers to the complete collection of the absolute time points in the whole event set, and $\tau$ denotes the relative time. Note that $\tau = 0$ indicates the synchrony of body and head event occurrences.

For example, a TR could be: Whenever someone orders dishes in a restaurant, he or she will have meals in around ten minutes, where $\tau = 10$.

**Metrics for TR Validity**    Since real-world data often contain noise, we introduce an adaptive $\triangle = [\tau(1 - \eta), \tau(1 + \eta)]$ to replace $\tau$ when evaluating statistically learned temporal regularities from events. For an evaluated TR abbreviated as $tr : (ev_b, ev_h, \tau, \eta)$, if an event $q : (ev, t)$ satisfies that $ev = ev_b$, we denote it as $b(q; tr)$; if $ev = ev_h$, we denote it as $h(q; tr)$. We define the support of a TR as the number of event pairs respectively satisfying the body and head:

$$sp(tr) = n(b(q; tr) \wedge h(q'; tr) \wedge (t' - t) \in \triangle(\tau, \eta)) \tag{2}$$

$\wedge$ denotes 'and', $\in$ denotes 'in', and $q : (ev, t), q' : (ev', t')$ refer to arbitrary two different events in the event set. Note that when calculating $sp(tr)$, we can only count one event once and in one pair to avoid overcounting when events occur in consecutive periods.

We respectively define the standard confidence, head coverage, and general confidence of a TR as:

$$sc(tr) = \frac{sp(tr)}{n(b(tr))}, \quad hc(tr) = \frac{sp(tr)}{n(h(tr))}, \quad gc(tr) = \frac{2}{\frac{1}{sc(tr)} + \frac{1}{hc(tr)}} \tag{3}$$

$n(b(tr)), n(h(tr))$ respectively represent the number of events $q : (ev, t)$ satisfying $ev = ev_b, ev = ev_h$ in the event set. Here we borrow the metrics $sc, hc, gc$ generally used in the rule mining field (Galárraga et al., 2015) to ensure fair and reasonable evaluations. We modify them by introducing an adaptive $\tau$ with $\eta$ to evaluate TR validity. Intuitively, standard confidence $sc$ can be viewed as the probability that the head event will occur within time $t + \triangle$ once a body event occurs at time $t$, whose statistical sufficiency is supported by $sp$. $hc$ and $gc$ can be interpreted similarly.

Above some $sp$, the higher the $gc$, the more valid a TR is. For a potential TR $tr : (ev_b, ev_h, \tau, \eta)$ with fixed event types $ev_b, ev_h$ and ratio $\eta$, its general confidence can be written as a function of $\tau$: $gc(\tau)$.

## 2.2 Tasks

For a fixed $\eta$ in $\triangle$, a potential TR can be an arbitrary $(ev_i, ev_j, \tau)$, where $i, j \in \mathbb{P}, \tau \in \mathbb{T}_r$ ($\mathbb{P}$ is the set of event types, $\mathbb{T}_r$ is the set of relative time points). Therefore, we define the two complementary tasks below to comprehensively evaluate the TR learning capabilities of event embeddings.

**TR Detection**    For a query $(ev_b, ev_h)$, its ground truth confidence $gc_g = \max_{\tau \in \mathbb{T}_r} gc(\tau)$. Queries whose $gc_g \geq \theta$ are considered to imply valid TRs that reveal good regularities, while queries whose $gc_g < \theta$ are considered to imply invalid TRs, where $\theta$ is a fixed threshold.

The task of TR detection is to identify valid TRs from all tested ones. The model is expected to determine whether a query $(ev_b, ev_h)$ implies a valid TR. The F1 score of all judgments is reported.

**TR Query**    Only queries that imply valid TRs are considered for testing. The ground truth relative time $\tau_g$ is set as what maximizes $gc(\tau)$ in computing $gc_g$. The model outputs $\tau'$.

The task of TR query is to output the correct $\tau' = \tau_g$ for valid TRs. For each tested query $(ev_b, ev_h)$, a ratio $r' = \frac{gc(\tau')}{gc_g}$ is computed. The averaged ratio $r$ of all queries is reported.

# 3  Noether Embedding

## 3.1  Inspirations from Noether's Theorem

**Noether's theorem**    In 1915, mathematician Emmy Noether proved one of the most fundamental theorems in theoretical physics: every differentiable symmetry of the action of a physical system with conservative forces has a corresponding conservation law. Specifically, time-translation symmetry corresponds to energy conservation.

**TRs indicate time-translation symmetries**    An event pair $(ev_b, t) \rightarrow (ev_h, t + \tau)$ can be regarded as a mapping of the body and the head event type over $t$ with a parameter $\tau$. Therefore, ideal TRs indicate the invariance of such mappings under the transformation of time translation since the mapping holds $\forall t \in \mathbb{T}_a$ for TRs.

**Construct embeddings with conserved local energies** Denote $\boldsymbol{q}(t; ev)$ as the embedding of each event sample, and $g(\boldsymbol{q}(t; ev_b), \boldsymbol{q}(t + \tau; ev_h))$ as the local energy of a corresponding body-and-head event pair of a potential TR. If $g$ is innately conserved, meaning that $g = g(\tau; ev_b, ev_h)$ is invariant to $t$, it indicates time-translation symmetry. We can then use the value of $g$ to approximate TR validity after training each event embedding $\boldsymbol{q}(t; ev)$. A more strict correspondence between NE variables and those in a physical system is shown in Appendix A.1.1.

**Noether-Inspired structural biases** Accordingly, the enabling factors of NE can be summarized as follows: (i) the event embedding $\boldsymbol{q}$ should be constructed to make each local energy $g$ remain invariant to $t$; (ii) the training loss should be designed to make the value of $g$ approximate TR validity; (iii) the local energy $g$ should be used as the decoding function. We thus construct NE as below.

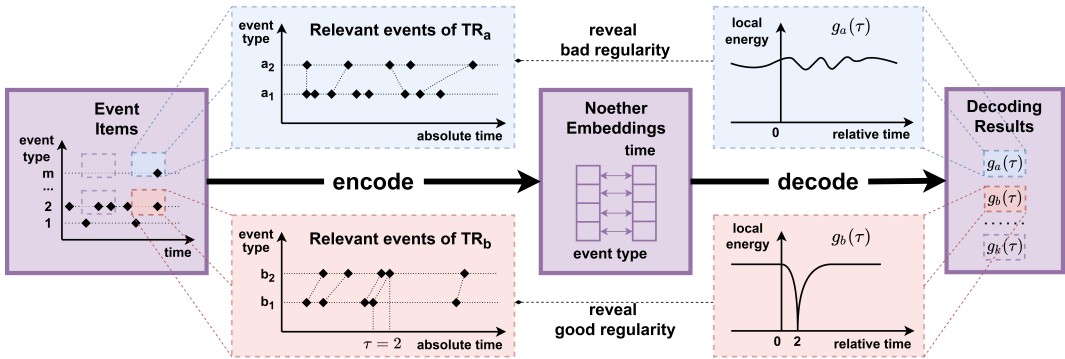

Figure 2: Illustration of NE. Solid lines and purple graphs in the middle jointly represent the data flow of NE. The red graphs below and the blue ones above demonstrate cases of a TR with significant temporal regularities and a TR with no. It is shown that each decoding result reveals an integrated temporal association of its relevant event pairs separated in time.

## 3.2 NE's Framework and Formulas

**Framework** As illustrated in Figure 2, NE uses a distributed storage of complex vectors to learn TRs. At the encoding stage, the event items are converted to NE representations through embedding each event sample, where TR validity is automatically calculated and stored in the embedding space. At the decoding stage, given each query $(ev_b, ev_h)$, potential TRs are detected and queried by directly calculating their relevant embeddings to derive the corresponding decoding results $g(\tau)$.

**The Encoding Stage**

$$\boldsymbol{q}(t; ev) = \boldsymbol{u}(ev) \circ \boldsymbol{r}(t) \tag{4}$$

In the event embedding above, $\circ$ denotes the Hadmard (or element-wise) product. $\boldsymbol{u}(ev), \boldsymbol{r}(t) \in \mathbb{C}^d$ are complex vectors where $\boldsymbol{u}(ev) = \frac{\boldsymbol{v}(ev)}{||\boldsymbol{v}(ev)||}$, $\boldsymbol{v}(ev) \in \mathbb{C}^d$, $\boldsymbol{r}(t) = e^{i\boldsymbol{\omega} t}$, $\boldsymbol{\omega} \in \mathbb{R}^d$, and $d$ is the dimension of vectors. Each event type $ev$ corresponds to an independently trainable vector of $\boldsymbol{u}(ev)$, while $\boldsymbol{\omega}$ is a global time vector for $\boldsymbol{r}(t)$ of all event embeddings. Note that the $d$ $\omega$s in $\boldsymbol{\omega}$ are fixed to a certain distribution. The event embedding $\boldsymbol{q}(t; ev)$ can thus be depicted as a rotation of event-type vectors $\boldsymbol{u}(ev)$ by time $\boldsymbol{r}(t)$ in the d-dimensional complex space.

The score function and loss function of each event sample are defined as follows:

$$f(t; ev) = \sum_{i=1}^{d} Real(\boldsymbol{q}(t; ev))_i \tag{5}$$

$$L(\xi; C_p, C_n) = (\frac{1}{\sqrt{d}}f(\xi) - C_p)^2 + \frac{1}{N}\sum(\frac{1}{\sqrt{d}}f(\xi') - C_n)^2 \tag{6}$$

We denote $\xi$ as a positive event sample and $\xi'$s as its generated negative samples whose number is $N$. For a positive sample $\xi : (ev, t)$, its negative samples $\xi'$s are the whole set of $\{(ev, t' \neq t)\}$

where $t' \in \mathbb{T}_a$. $C_p, C_n$ are two different constants for positive and negative samples, respectively. $C_p, C_n \in [-1, 1]$ because $||\boldsymbol{u}(ev)|| = 1$, and we generally set $C_p = 1, C_n = 0$.

Until the training converges, events and TRs form a distributed storage, which includes the global time vector of $\boldsymbol{\omega}$ and trainable event type vectors $\boldsymbol{u}(ev)$. At the decoding stage, no training but only the inference of vectors is conducted, as described below.

**The Decoding Stage**    The decoding function for a query $(ev_b, ev_h)$ is:

$$g(\tau) = ||\boldsymbol{u}_b - \boldsymbol{u}_h \circ \boldsymbol{r}(\tau)||^2 \tag{7}$$

$\boldsymbol{u}_b, \boldsymbol{u}_h$ are the event type vectors respectively of $ev_b, ev_h$. $\tau$ is traversed through set $\mathbb{T}_r$ of the relative time points such as $\mathbb{T}_r : \{-\tau_{max}, ..., 0, ..., \tau_{max}\}$ to plot the decoding results. $\min_{\tau \in \mathbb{T}_r} g(\tau)$ is computed, which is compared with a global threshold $g_{th}$ to decide whether a potential TR is valid or not (for TR detection). For a valid TR, the $\tau'$ which minimizes $g(\tau), \tau \in \mathbb{T}_r$ is selected as the model output of the relative time (for TR query).

Since $\boldsymbol{r}(t + \tau) = \boldsymbol{r}(t) \circ \boldsymbol{r}(\tau)$, $\boldsymbol{r}(t) \circ \overline{\boldsymbol{r}(t)} = \mathbf{1}, \forall t \in \mathbb{T}_a$, the decoding function exactly indicates the conserved local energy: $g(\tau) = ||\boldsymbol{u}_b - \boldsymbol{u}_h \circ \boldsymbol{r}(\tau)||^2 = ||\boldsymbol{u}_b \circ \boldsymbol{r}(t) - \boldsymbol{u}_h \circ \boldsymbol{r}(t + \tau)||^2 = ||\boldsymbol{q}_b(t) - \boldsymbol{q}_h(t + \tau)||^2, \forall t \in \mathbb{T}_a$. This indicates that $g = g(\tau; ev_b, ev_h)$ is invariant to $t$, and the conserved energy $g$ of arbitrary two event samples is of a quadratic form in the embedding space.

## 3.3    Why NE is Efficient

Briefly speaking, the Fourier-like representations enable NE's large-capacity storage for TR validity and event occurrences, serving as a prerequisite for NE to learn TR effectively. The Noether-inspired structural biases further leads to NE's efficient TR learning capabilities. Here we only illustrate the effect of the structural biases. The reasons why NE learns TR effectively is explained in Appendix A.2.

**Data-efficiency by Encoding Translation Symmetries**    The invariance of $g(\tau)$ to $t$ means that the value of $g(\tau)$ is determined by a competitive effect between sample pairs across time. Considering event sample pairs $(ev_b, t), (ev_h, t + \tau)$ with varying $t$ in the embedding space, if a sample pair are both positive or both negative samples, they will decrease $g(\tau)$ since their score functions are mapped to the same constant. Otherwise, they will increase $g(\tau)$. Since $g(\tau)$ is invariant to $t$, $g(\tau)$ is trained to balance these two forces. Therefore, the value of $g(\tau)$ after training convergence is generally determined by the ratio of sample pairs with increasing or decreasing forces. $g(\tau)$ is thus insensitive to the number of sample pairs that are both positive. This results in a data-efficient TR formation in NE, even with limited event occurrences from which to learn a TR.

**Time-efficiency by Decoding Conserved Energies**    By calculating $g(\tau) = ||\boldsymbol{u}_b - \boldsymbol{u}_h \circ \boldsymbol{r}(\tau)||^2, \tau \in \mathbb{T}_r$, we enable efficient TR querying for each query $(ev_b, ev_h)$. This process has a constant time complexity since $\mathbb{T}_r$ is an arbitrary user-selected set of relative time points, and the vector dimension $d$ can be effectively handled using GPUs. Importantly, the querying time is independent of the number of events in the entire event set and the relevant event occurrences supporting the queried TR.

## 4    Experiment

It is important to highlight that in our evaluation, we initially compare NE and classic embeddings in terms of learning effectiveness (Section 4.2), without considering the efficiency requirements. As classic embeddings are shown to be ineffective in learning TRs, we then focus on demonstrating the learning efficiency of NE in Section 4.3.

### 4.1    Experimental Setting

**Dataset**    A temporal knowledge graph (TKG) comprises $(s, p, o, t)$ quadruples (Leblay & Chekol, 2018), where $s, p, o, t$ represent the subject, predicate, object, and time. TKG is widely used in a variety of fields to represent global political events (Trivedi et al., 2017), financial incidents (Yang et al., 2019), user-item behaviors (Xiao et al., 2020), etc. Notably, ICEWS (Boschee et al., 2015) and GDELT (Leetaru & Schrodt, 2013) are two popular data sources for TKG research (Cai et al., 2022).

In our experiments, we use ICEWS14 and ICEWS18, the same as in (Han et al., 2020). They contain global political events in 2014 and 2018, and we denote them as D14 and D18, respectively. We also use the GDELT released by (Jin et al., 2019), which contains global social events from 2018/1/1 to 2018/1/31. In the experiments, we denote each $(s, p, o)$ triple as a specific event type $ev$. It is worth mentioning that alternative settings, such as representing each predicate $p$ as an event type $ev$, are also applicable to our model.

**Model Implementation**    For NE, $d = 400, C_p = 1, C_n = 0$ and the global time vector $\boldsymbol{\omega}$ is set as $\omega_k = \frac{(2\pi \times \omega_{max})^{\frac{k}{d}-1}}{T_a}, k = 0, 1, ..., d - 1$, where $T_a$ is the number of absolute time points, and $\omega_{max}$ is a tunable hyperparameter set as 600. The training details are in Appendix B.1. We compare NE with six classic and vastly different TKG embeddings, DE-SimplE (Goel et al., 2020), TeRo (Xu et al., 2020b), ATiSE (Xu et al., 2020a), TNTComplex (Lacroix et al., 2020), BoxTE (Messner et al., 2022), and TASTER (Wang et al., 2023) with their original parameter settings and $d = 400$ the same as NE. We set the queried time set $\mathbb{T}_r = \{1 - T_a, ..., 0, ..., T_a - 1\}$ at the decoding stage. In this way, only one query needs to be decoded between each $(ev_i, ev_j)$ and $(ev_j, ev_i)$, $i \neq j$.

Adaptations. All baselines can only output their score function $f'(t)$ to decode event occurrences but cannot directly decode TR validity by $g(\tau)$ as NE does. For comparison, we add an interface $g'(\tau) = \frac{\sum_{t,t+\tau \in \mathbb{T}_a} f'_b(t) f'_h(t+\tau)}{\sum_{t \in \mathbb{T}_a} f'_b(t) \cdot \sum_{t \in \mathbb{T}_a} f'_h(t)}$ to these models that indirectly compute TR validity from the decoded event occurrences. We also evaluate NE with $g'(\tau)$ to show the validity of $g'(\tau)$ itself.

**Evaluation Details**    We select $(ev_b, ev_h)$s for tests whose event occurrences of $ev_b$ and $ev_h$ are both $\geq 2$. Otherwise, their supports ($sp$ in Definition 2) will be too small for evaluating TR validity. We set $\eta = 0.1$ in $\triangle$s for strict evaluations and take the upper integer $\triangle = [\tau - \lceil \tau\eta \rceil, \tau + \lceil \tau\eta \rceil]$. Note that in the extreme situation where body and head event occurrences both $= 2$, stochastic noises are still quite unlikely to interfere with the evaluation of TR validity since $\eta = 0.1$ is strict. Only forward or reverse queries $((ev_b, ev_h)$s whose $s_b = s_h, o_b = o_h$ or $s_b = o_h, o_b = s_h$ for $(s, p, o, t)$ quadruples) are tested for better interpretability without sacrificing generality. We set $\theta = 0.8$ to distinguish between valid and invalid TRs. The fact that TRs whose $gc_g \sim 0.8$ are of a tiny percentage of all tested TRs adds to the rationality of such metrics. Ablations where $\theta = 0.7, 0.9$ are in Appendix B.3.3.

In comparative studies with baselines 4.2, we report the highest F1 in TR detection by tuning the global threshold $g_{th}$ (defined in Section 3.2) after embedding the whole event set to achieve full evaluations. We also remove TRs whose $\tau_g = 0$ for TR query because they account for most valid TRs but can hardly reflect the query difficulty. In NE's demonstration studies 4.3 4.4, we first use D14 to derive the global threshold $g_{th}$ with the highest F1 in TR detection and then apply the same $g_{th}$ for evaluating NE's performance in D18. This setting better demonstrates NE's practicality.

## 4.2    Comparisons of Learning Effectiveness

Table 1: Statistical results on ICEWS14, ICEWS18, and GDELT

| Embedding | TR Detection (F1) | | | TR Query (r) | | |
|---|---|---|---|---|---|---|
| | D14 | D18 | GDELT | D14 | D18 | GDELT |
| TNTComplEx | 0.26 | 0.18 | 0.08 | 0.08 | 0.08 | 0.01 |
| DE-SimplE | 0.22 | 0.20 | - | 0.09 | 0.09 | - |
| TASTER | 0.18 | 0.15 | 0.08 | 0.09 | 0.09 | 0.00 |
| TeRo | 0.43 | 0.64 | 0.16 | 0.08 | 0.08 | 0.01 |
| BoxTE | 0.40 | 0.40 | 0.18 | 0.08 | 0.08 | 0.01 |
| ATISE | 0.40 | 0.44 | 0.18 | 0.08 | 0.08 | 0.01 |
| NE with $g'(\tau)$ | 0.78 | 0.79 | 0.48 | 0.85 | 0.83 | 0.83 |
| NE with $g(\tau)$ | **0.82** | **0.83** | **0.51** | **0.87** | **0.86** | **0.85** |

**Performances**    Table 10 shows that NE with $g(\tau)$ has an overwhelming advantage over all baselines with $g'(\tau)$, both in detecting valid TRs and querying the relative time on all evaluated datasets. The

excellent performance of NE with $g'(\tau)$ indicates that $g'(\tau)$ itself is valid and thus guarantees fair comparisons. It is worth noting that NE is intrinsically different from all existing baselines because only NE can directly decode TR validity by $g(\tau)$. In contrast, baselines can only decode event occurrences $f'(t)$ from which to indirectly calculate TR validity (such as by $g'(\tau)$). Detailed results of precision and recall rates with error bars are reported in Appendix B.2.

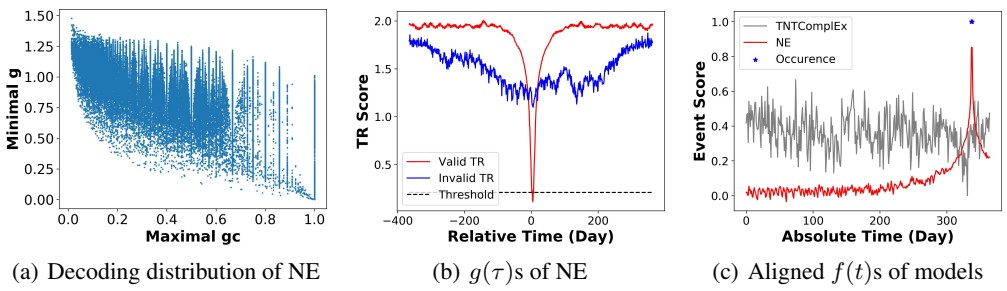

(a) Decoding distribution of NE        (b) $g(\tau)$s of NE        (c) Aligned $f(t)$s of models

Figure 3: Illustrations of NE or baselines

**Discussion**    Figure 3(a) shows NE's decoding distribution $g_m = \min_{\tau \in \mathbb{T}_r} g(\tau)$ by each query's ground truth $gc_g = \max_{\tau \in \mathbb{T}_r} gc(\tau)$. It can be observed that the decoded conserved local energy accurately reveals the TR validity, which enables NE to successfully distinguish between valid and invalid TRs, as demonstrated in the case shown in Figure 3(b). Table 10 shows that baselines with $g'(\tau)$ still perform poorly. This is mainly because their $f(t)$s do not fill well. Specifically, Figure 3(c) illustrates that TNTComplEx has much noise in its $f'(t)$ compared to NE. The reason is that baseline models are generally designed to achieve good performance in the completion task and, therefore, over-apply the generalization capabilities of distributed representations, which hinders the fit of event occurrences.

## 4.3 NE's Superior Learning Capabilities

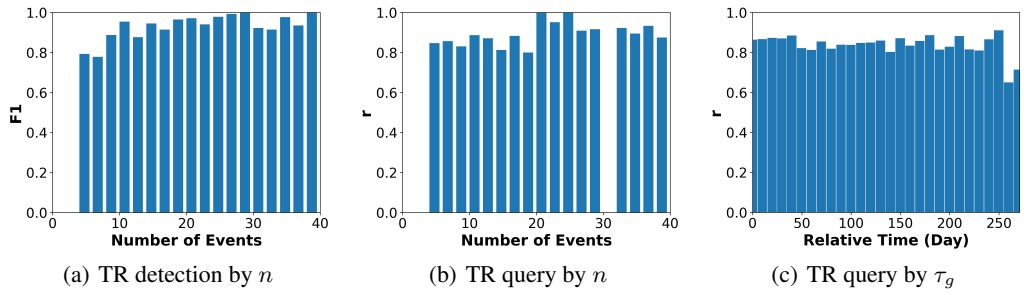

(a) TR detection by $n$        (b) TR query by $n$        (c) TR query by $\tau_g$

Figure 4: Grouped performances of NE

**Data Efficiency**    In Figure 4(a) and 4(b), we group TRs by their number $n$ of relevant events. It is shown that NE accurately detects valid TRs and reports correct $\tau$s with only two event pairs as positive samples. This performance is comparable to humans, able to form temporally associative memories with minimal experience (Hudson et al., 1992; Bauer & Mandler, 1989; Schapiro et al., 2017). Note that the maximum group in the test has $n > 400$, while we only show groups with $n \le 40$ for display considerations.

**Time Efficiency**    As explained in 3.3, NE's specific decoding function $g(\tau)$ enables NE to retrieve TRs in a constant time complexity by vector computations. Calculating $g'(\tau)$ of classic embeddings, however, requires an additional time complexity relevant to $T_a$ (the number of absolute time points).

**Storage Efficiency**    In addition to the data-efficient and time-efficient properties, NE is, in fact, also a storage-efficient memory for TRs and event occurrences. Here is a detailed analysis:

The storage of NE vectors, denoted as $S(NE)$, can be calculated as follows: $S(NE) = S(ev-vector) + S(time-vector) = 2*N*d*64bit + 2*d*64bit$. In our experiments, we used torch.LongTensor and N represents the number of event-type vectors. On the other hand, the storage of exact counting, denoted as $S(CT)$, can be calculated as follows: $S(CT) = S(TR) + S(event) = N^2 * T_a * log_2(n/N)bit + N*(n/N)*log_2(T_a)bit$. Here, $n$ represents the number of all event occurrences. We reserved the storage accuracy of TR validity to effectively distinguish different values, resulting in approximately $log_2(n/N)bit$ for each TR validity $(ev_b, ev_h, \tau)$.

For the ICEWS14 and ICEWS18 datasets, where $d = 400, T_a = 365$, and $n = 90730, 468558, N = 50295, 266631$, we calculated the compression ratio $\frac{S(CT)}{S(NE)}$ of NE as 421 and 2336, respectively. This remarkable capability of NE can be attributed to the fact that it separately stores the information of TR validities $(ev_b, ev_h, \tau)$ using event-type vectors and a global time vector. By representing the common information of related TRs efficiently in memory, NE achieves a compression ratio that is approximately linear to the number of event types.

**Flexibility**  In Figure 4(c), we group valid TRs by their golden $\tau_g$s. NE is shown to be flexible for learning TRs with $\tau$s varying broadly, comparable to humans with stable memory codes for various time intervals in the hippocampus (Mankin et al., 2012).

## 4.4 NE's Wide Potential Use

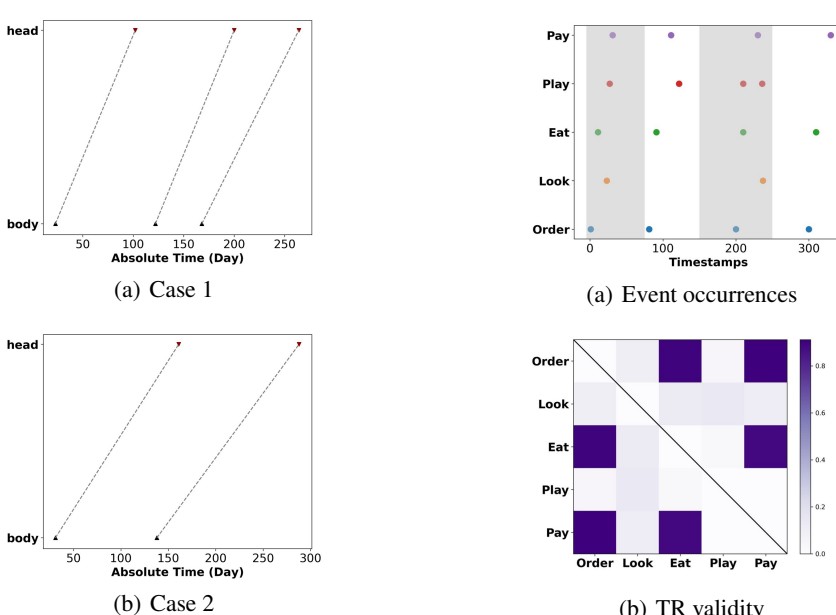

(a) Case 1

(b) Case 2

Figure 5: Social event prediction

(a) Event occurrences

(b) TR validity

Figure 6: Personal decision making

**Potential Use in Social Event Prediction**  In D18, NE successfully reports 21010 valid TRs with an F1 score of 0.83. The encoding stage takes around 1 hour, while decoding only takes less than 1 minute. Cases are presented below and in Figure 5, with additional cases available in Appendix B.4.

(1) Case 1. Citizen (India) will *Reject* to Narendra Modi (events in day 23, 122, and 168) in around 87 days whenever Narendra Modi *Appeal for diplomatic cooperation (such as policy support)* to Citizen (India) (events in day 102, 200, and 264).

(2) Case 2. Russia will *Meet at a 'third' location* with Ukraine (events in day 31 and 138) in around 136 days whenever Ukraine *Use conventional military force* to Russia (events in day 161 and 288).

Since the TRs mined can generalize across time, the results above imply NE's potential use in both reliable and interpretable event predictions urgently needed in the big data era (Zhao, 2021).

**Potential Use in Personal Decision Making**   Consider an intelligent machine that has visited a restaurant four times, with the occurrence time of each event episode used as input for NE, as shown in Figure 6(a). After training all events, the decoded TR validity $\min_{\tau \in \mathbb{T}_r} g(\tau)$ is transformed linearly and demonstrated in Figure 6(b). Despite the recurrent TRs on the slash that can be set aside, valid TRs such as 'order dishes –(about 10 minutes)–> have meals' are well distinguished from invalid ones such as 'order dishes –(about 5 minutes) –> look at the floor'.

Combining NE with front-end methods that take unstructured videos as input and output event items in the form of $(ev, t)$, and with back-end methods that use the decoded valid TRs to guide decision-making, NE has the potential to aid intelligent machines in surviving in changing environments by generalizing from little experience, just as human beings (Goyal & Bengio, 2022; Xue et al., 2018).

**Potential Use in Memory-constrained Scenarios**   As discussed in 4.3, NE approximately reduces the required storage space from $M^2$ to $M$ in our experimental settings, where $M$ is the number of event types. Therefore, NE holds significant potential for applications in memory-constrained scenarios like the edge. This is important when $M$ is large, which is usual in the big-data era.

## 4.5   Ablation Studies

Here we demonstrate ablation studies of loss constants $C_p, C_n$ and time vector $\boldsymbol{\omega}$, while those for dimension $d$ and event type vector $\boldsymbol{u}$ are shown in Appendix B.3.

Table 2: NE on ICEWS14 in different $C_p$ and $C_n$ settings

| $C_n$ / $C_p$ | TR Detection (F1) | | | | | TR Query (r) | | | | |
|---|---|---|---|---|---|---|---|---|---|---|
| | 0.4 | 0.2 | 0 | -0.2 | -0.4 | 0.4 | 0.2 | 0 | -0.2 | -0.4 |
| 1 | 0.78 | 0.80 | 0.82 | 0.81 | 0.80 | 0.86 | 0.87 | 0.87 | 0.87 | 0.87 |
| 0.8 | 0.64 | 0.67 | 0.79 | 0.80 | 0.80 | 0.85 | 0.86 | 0.86 | 0.87 | 0.86 |
| 0.6 | 0.29 | 0.40 | 0.68 | 0.79 | 0.79 | 0.44 | 0.82 | 0.86 | 0.86 | 0.85 |
| 0.4 | 0.18 | 0.31 | 0.49 | 0.76 | 0.79 | 0.12 | 0.35 | 0.84 | 0.84 | 0.81 |
| 0.2 | 0.53 | 0.19 | 0.27 | 0.71 | 0.78 | 0.27 | 0.05 | 0.28 | 0.79 | 0.71 |

**Loss Constants**   Table 2 shows that NE performs optimally when $C_p = 1$ and $C_n = 0$. In fact, as $C_p$ approaches 1, the $g(\tau)$ of perfect TRs ($gc(\tau) = 1, \eta = 1$) is enforced to converge to its minimum 0. This global constant for all potential TRs in the embedding space allows $g(\tau)$ to reveal TR validity better. In terms of $C_n$, setting it to 0 results in negative samples occupying the largest embedding space. Since negative samples comprise most of all trained event samples, this setting improves the fit of negative samples and optimizes NE's performance.

Table 3: NE on GDELT with different $\omega_{max}$s

| $\omega_{max}$ | 1 | 5 | 10 | 50 | 100 | 200 | 400 | 600 | 800 |
|---|---|---|---|---|---|---|---|---|---|
| TR Query (r) | 0.15 | 0.93 | 0.92 | 0.85 | 0.85 | 0.85 | 0.85 | 0.85 | 0.85 |
| TR Detection (F1) | 0.22 | 0.45 | 0.55 | 0.56 | 0.54 | 0.53 | 0.53 | 0.51 | 0.53 |

**Maximal Frequency Coefficient**   Table 3 shows that NE performs optimally with different values of $\omega_{max}$, respectively, in the TR detection and query task.

Table 4: NE on the three datasets with increasing events, and with different distributions of $\{\omega_k\}$

| $\{\omega_k\}$ | D14 (90730 events) | | D18 (468558 events) | | GDELT (2278405 events) | |
|---|---|---|---|---|---|---|
| | linear | exponential | linear | exponential | linear | exponential |
| TR Query (r) | 0.81 | 0.82 | 0.75 | 0.82 | 0.24 | 0.85 |

**Frequency Distribution**    Table 4 shows that the larger the dataset, the more exponential distribution ($\omega_k = \frac{(2\pi \times \omega_{max})^{\frac{k}{d}} - 1}{T_a}, k = 0, 1, ..., d-1$) surpasses linear distribution ($\omega_k = \frac{2\pi \times k \times \omega_{max}}{d \times T_a}, k = 0, 1, ..., d-1$) with the same parameters of $d = 400, \omega_{max} = 600$. This suggests that real-world event occurrences depend more on low-frequency terms than high-frequency ones.

## 5    Related Work

**Event Schema Induction**    In the natural language processing (NLP) field, a significant research focus is on inducing event schemas from text (Huang et al., 2016; Li et al., 2021), including from language models (Dror et al., 2022), to support downstream applications such as search, question-answering, and recommendation (Guan et al., 2022). These NLP methods aim to organize known event regularities already given as priors for the extracting algorithm (such as extracting 'earthquake -> tsunami' from the sentence 'An earthquake causes a tsunami.') and focus on the schemas for use. In contrast, our tasks are designed to learn event regularities directly from experience without supervision. Specifically, the only prior models know is whether an event occurs, and models are required to detect valid TRs from all potential ones and report the correct relative time of valid TRs.

**Temporal Rule Mining**    Various temporal rules are mined from event sets to reveal regularities in industry, security, healthcare, etc (Segura-Delgado et al., 2020; Chen et al., 2007; Yoo & Shekhar, 2008; Namaki et al., 2017). Although the search methods used discover event regularities directly from events without supervision, both the mined rules and source events are generally stored as symbolic representations in list form. In contrast, by applying event embeddings, NE is a distributed and approximate memory for both TRs and event items. NE strikes a balance between storage efficiency and storage accuracy compared to exact counting, as detailedly discussed in 4.3.

**Embedding Models of Structured Data**    Within all embedding models of static and temporal knowledge graphs (Chen et al., 2020; Cai et al., 2022; Wang et al., 2020; Messner et al., 2022), three are most related to NE. RotatE (Sun et al., 2019) represents each entity and relationship as a complex vector to model relation patterns on knowledge graphs, and TeRo (Xu et al., 2020b) represents time as a rotation of entities to model time relation patterns on temporal knowledge graphs. While both RotatE and TeRo introduce complex vectors for better completion performance, NE first explores using complex vectors for TR detection in events. In particular, the specific use of complex vectors in RotatE encodes inverse relations and in TeRo encodes asymmetric and reflexive relations. NE, instead, apply rotating complex unit vectors to encode time-translation symmetries of all potential TRs. IterE (Zhang et al., 2019) construct a decodable embedding model to discover rules for better knowledge graph completion performance. While we take functional inspiration from IterE that embedding models can jointly encode data and decode regularities, we focus on event data and define the new problems of TR detection and TR query. Specifically, while IterE focuses on discrete variables, NE focuses on the continuous variable of time that involves Fourier-like transformations.

To summarize, TR detection and TR query focus on achieving human-like schema learning capabilities rather than pursuing better support for NLP applications like the event schema induction task. Meanwhile, NE leverages the advantages of distributed representations over symbolic ones of search methods in temporal rule mining and is distinct from existing embedding models of structured data.

## 6    Conclusion

We have developed NE which for the first time enables data-efficient TR formation and time-efficient TR retrieval simply through embedding event samples. We have formally defined the tasks of TR detection and TR query to comprehensively evaluate the TR learning capabilities of embedding models. We have demonstrated NE's potential use in social event prediction, personal decision-making, and memory-constrained scenarios. We hope that we have facilitated the development of human-like event intelligence.

One limitation of NE is that when the vector dimension $d$ is set much lower than the number of absolute time points $T_a$, significant performance degradation of NE will occur as observed in the GDELT experiment. Future research is needed to improve this weakness. The privacy issues potentially brought about by TR detection and the causality of TRs should also be handled properly.

## Acknowledgements

We sincerely thank Rong Zhao, Hao Zheng, Dahu Feng, Lukai Li, Hui Zeng, Wenhao Zhou, Yu Du, Songchen Ma, and Faqiang Liu for their valuable comments and encouragements. This work was supported by the National Nature Science Foundation of China (nos. 62088102, 61836004) and the National Key Research and Development Program of China (no. 2021ZD0200300).

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

# Appendix

## A  Theoretical Illustrations

### A.1  Interdisciplinary Correspondences

#### A.1.1  Physical Correspondence

Considering each event type as a particle, the event embedding $\boldsymbol{q}(t; ev)$ can be viewed as its generalized coordinate at time $t$ in the d-dimensional complex embedding space. Suppose that each two particles $ev_b, ev_h$ are connected by a spring with the force linear to their distance, their potential energy can then be expressed as $||\boldsymbol{q}_b(t) - \boldsymbol{q}_h(t + \tau)||^2$, where $\tau$ is a parameter. As $\boldsymbol{q}(t; ev)$ changes with time, its kinetic energy also changes and can be viewed as being driven by external nonconservative forces. Using the extended Noether's theorem, we can see that the work of external forces and the kinetic energy offset, leading to a conserved quantity of time-translation symmetry that only includes the potential energy instead of the total energy. The inductive bias of Noether Embedding exactly enforces that such local potential energies are innately conserved, which means that $||\boldsymbol{q}_m(t_k) - \boldsymbol{q}_n(t_k + \tau_r)||^2 = ||\boldsymbol{q}_m(t_s) - \boldsymbol{q}_n(t_s + \tau_r)||^2, \forall ev_m, ev_n \in \mathbb{P}, t_k, t_s \in \mathbb{T}_a, \tau_r \in \mathbb{T}_r$, where $\mathbb{P}, \mathbb{T}_a, \mathbb{T}_r$ refer to the set of event types, absolute time points, and relative time points, respectively.

#### A.1.2  Biological Correspondence

The formation of TRs relies on two central functions, namely measurement and integration. The measurement function involves comparing the temporal distance between each two events. It is claimed that Laplace transformation exists in the hippocampus for representing time (Howard et al., 2014). Specifically, the population of temporal context cells (or referred to as ramping cells (Tsao et al., 2018)) in the lateral entorhinal cortex is discovered to code time with a variety of rate constants (Bright et al., 2020). $\boldsymbol{r}(t)$ in NE functionally corresponds to such a cell population, which stores the rate distribution in vector $\boldsymbol{\omega}$ of $\boldsymbol{r}(t)$. The integration function aggregates event pairs separated in time to form TRs. Statistical learning for schema formation is reported to occur in the pathway connecting EC to CA1 in the hippocampus (Schapiro et al., 2017). In NE, we achieve the same function through the time-translation symmetry introduced by $\boldsymbol{r}(t)$: $\boldsymbol{r}(t + \tau) = \boldsymbol{r}(t) \circ \boldsymbol{r}(\tau)$, $\boldsymbol{r}(t) \circ \overline{\boldsymbol{r}(t)} = \mathbf{1}, \forall t \in \mathbb{T}_a$.

### A.2  Revelation of TR Validity

#### A.2.1  Effectiveness in TR Query

NE is an effective method for TR query due to its distributed storage of cross-correlations.

The score function $f(t)$ describes the occurrences of each event type, where its value is mapped to $C_p$ for time points implying event occurrences and $C_n$ for those that do not. The support $sp(\tau)$ of two event types $ev_b, ev_h$ can be then approximated by calculating the time-lagged cross-correlation $R_{f_b, f_h}(\tau) = \sum_{t, t+\tau \in \mathbb{T}_a} f_b(t) f_h(t + \tau)$ after training convergence.

Denote $F(\omega)$ as the Fourier expansion of $f(t)$. Awaring that in NE, $f(t) = \sum_{j=1}^{d} Real(\boldsymbol{u} \circ e^{i\boldsymbol{\omega} t})_j$, we can see that the encoding stage enables a $d$-dimensional Fourier-like expansion for each $f(t)$. The time vector $\boldsymbol{\omega}$ provides the expansion basis and $\boldsymbol{u}$ stores the coefficients as $F(\omega)$s.

By Fourier expansions, the corresponding correlation $R'_{f_b, f_h}(\tau) = \int_{-\infty}^{+\infty} f_b(t) f_h(t + \tau) dt = \int_{-\infty}^{+\infty} \overline{F_b(\omega)} F_h(\omega) e^{i\omega\tau} d\omega$. In NE, $g(\tau) = ||\boldsymbol{u}_b - \boldsymbol{u}_h \circ \boldsymbol{r}(\tau)||^2 = 2 - 2\sum_{j=1}^{d} Real(\overline{\boldsymbol{u}_b} \circ \boldsymbol{u}_h \circ e^{i\boldsymbol{\omega}\tau})_j$. Therefore, each $g(\tau)$ reveals exactly the fitted time-lagged cross-correlation and is thus proportional to the support $sp(\tau)$ of a TR. This guarantees NE's effectiveness in TR query.

#### A.2.2  Effectiveness in TR Detection

**Notations**  Within all event samples trained for a potential TR, suppose that $m$ event pairs share the same relative time of $\tau$, where $m_1$ pairs are both positive or both negative samples. In each remaining $m_2 = m - m_1$ pair, one is a positive sample while the other is a negative one. Denote that $\mathbb{M} = \{1, 2, ..., m\}, \mathbb{M}_1 = \{i_1, i_2, ..., i_{m_1}\}, \mathbb{M}_2 = \{j_1, j_2, ..., j_{m_2}\}$. There exists such least upper bound $\eta$ that $(\frac{1}{\sqrt{d}} f(t; ev) - C)^2 < \eta$ for the score functions of all $2m$ samples, where $C = C_p, C_n$.

Denote that $\boldsymbol{a}_i = (cos(\boldsymbol{\omega}_1 t_i), cos(\boldsymbol{\omega}_2 t_i), ..., cos(\boldsymbol{\omega}_d t_i), sin(\boldsymbol{\omega}_1 t_i), sin(\boldsymbol{\omega}_2 t_i), ..., sin(\boldsymbol{\omega}_d t_i))^T$, $i \in \mathbb{M}$, where $d$ is the dimension of NE vectors, $t_i$ is time of the $i$th body event in $m$ sample pairs. Denote that in the decoding function, $\boldsymbol{u}_b - \boldsymbol{u}_h \circ \boldsymbol{r}(\tau) = \boldsymbol{\alpha} - i\boldsymbol{\beta}$ where $\boldsymbol{\alpha}, \boldsymbol{\beta}$ are both real vectors, and $\boldsymbol{x} = (\boldsymbol{\alpha}_1, \boldsymbol{\alpha}_2, ..., \boldsymbol{\alpha}_d, \boldsymbol{\beta}_1, \boldsymbol{\beta}_2, ..., \boldsymbol{\beta}_d)^T$. Denote that $c_1 = \max\limits_{i \in \mathbb{M}_1}(cos(\boldsymbol{a}_i, \boldsymbol{x}))^2, c_2 = \min\limits_{j \in \mathbb{M}_2}(cos(\boldsymbol{a}_j, \boldsymbol{x}))^2$.

Using the trigonometric inequality, we can derive the following conclusions:

**Theorem 1**   Within the $m_1$ sample pairs, if $c_1 > 0$, then $g(\tau) < \frac{4\eta}{c_1}$.

**Theorem 2**   Within the $m_2$ sample pairs, if $c_2 > 0$ and $|C_p - C_n| > 2\sqrt{\eta}$, then $g(\tau) > \frac{(|C_p - C_n| - 2\sqrt{\eta})^2}{c_2}$.

**Implications**   $d, T_a, \boldsymbol{\omega}, C_p, C_n$ jointly result in the probability distributions of $c_1, c_2$, where $c_1, c_2 > 0$ is generally guaranteed by experimental settings. Two conclusions can then be drawn from the two theorems. (1) Convergence. We can see from theorem 1 that $g(\tau) \to 0$ as $\eta \to 0$. This implies that the $g(\tau)$ of perfect TRs ($gc(\tau) = 1, \eta = 1$) will converge to its minimum of 0. (2) Competition. Since $g(\tau)$ is invariant to $t$, comparing these two theorems also tells us about the competing effect of well-trained sample pairs for the value of $g(\tau)$, generally affected by the ratio $\frac{m_1}{m_2}$. These two conclusions, along with the fact that $g(\tau)$ is proportional to the support $sp(\tau)$ (as illustrated in A.2.1), jointly make $g(\tau)$ reveal the TR validity $gc(\tau)$ and thus guarantees NE's effectiveness in TR detection.

It is worth noting that $d$ is generally set as $d > T_a$ to control the values of $c_1, c_2$, where $T_a$ is the number of absolute time points of the whole event set. Otherwise, NE's TR detection performance will be interfered. For example, if $d \ll T_a$, then $d \ll m$ in most cases. It will thus be very likely that $c_1 = 0$ so that theorem 1 can not be applied in NE.

# B    Experimental Supplements

## B.1    Training Details

All models are trained for 100 epochs on each dataset using the Adagrad optimizer (with a learning rate of 0.01) and the StepLR learning rate scheduler (with a step size of 10 and gamma of 0.9). The experiments are conducted on a single GPU (GeForce RTX 3090). The hyper-parameters of NE are fine-tuned using a grid search to achieve relatively optimal results.

## B.2    Main Results with Error Bars

To ensure the reliability of the results, the experiments are repeated three times, and the error bars are derived accordingly. Table 5 and 6 show that the main results are quite stable with small error bars. Note that the precision and recall rates in Table 6 correspond exactly to the F1 scores in Table 5. The reason why the recall rate of NE is lower than that of TASTER is that we report the highest F1 score of each model in comparative studies by tuning their respective global threshold, denoted as $g_{th}$. As the F1 score is calculated as the harmonic mean of precision and recall rates, TASTER achieves its highest F1 score by reporting many false positives, resulting in a relatively high recall rate but an extremely low precision rate.

## B.3    Additional Ablations

Unless otherwise specified, the experiments below adopt the original parameter settings as described in the main text.

### B.3.1    Normalization of Event Type Vectors

Table 7 demonstrates that normalized event type vectors slightly outperform unnormalized ones in the TR detection task.

Table 5: TR detection by F1 scores and TR query by confidence ratios

| Embedding | TR Detection (F1) | | | TR Query (r) | | |
|---|---|---|---|---|---|---|
| | D14 | D18 | GDELT | D14 | D18 | GDELT |
| TNTComplEx | 0.26±0.01 | 0.18±0.00 | 0.08±0.01 | 0.08±0.00 | 0.08±0.00 | 0.01±0.00 |
| DE-SimplE | 0.22±0.00 | 0.20±0.00 | - | 0.09±0.00 | 0.09±0.00 | - |
| TASTER | 0.18±0.00 | 0.15±0.00 | 0.08±0.00 | 0.09±0.00 | 0.09±0.00 | 0.00±0.00 |
| TeRo | 0.43±0.02 | 0.64±0.01 | 0.16±0.00 | 0.08±0.00 | 0.08±0.00 | 0.01±0.00 |
| BoxTE | 0.40±0.01 | 0.40±0.02 | 0.18±0.01 | 0.08±0.00 | 0.08±0.00 | 0.01±0.00 |
| ATISE | 0.40±0.01 | 0.44±0.01 | 0.18±0.01 | 0.08±0.00 | 0.08±0.00 | 0.01±0.00 |
| NE with $g'(\tau)$ | 0.78±0.00 | 0.79±0.00 | 0.48±0.00 | 0.85±0.00 | 0.83±0.00 | 0.83±0.00 |
| NE with $g(\tau)$ | **0.82**±0.00 | **0.83**±0.00 | **0.51**±0.00 | **0.87**±0.00 | **0.86**±0.00 | **0.85**±0.00 |

Table 6: TR Detection by precision and recall rates

| Embedding | Precision | | | Recall | | |
|---|---|---|---|---|---|---|
| | D14 | D18 | GDELT | D14 | D18 | GDELT |
| TNTComplEx | 0.22±0.01 | 0.11±0.00 | 0.04±0.00 | 0.33±0.02 | 0.50±0.01 | 1.00±0.01 |
| DE-SimplE | 0.16±0.00 | 0.14±0.00 | - | 0.35±0.03 | 0.33±0.00 | - |
| TASTER | 0.10±0.00 | 0.08±0.00 | 0.04±0.00 | 0.99±0.01 | 0.99±0.01 | 0.97± 0.03 |
| TeRo | 0.51±0.04 | 0.91±0.01 | 0.20±0.04 | 0.37±0.02 | 0.49±0.00 | 0.14±0.02 |
| BoxTE | 0.40±0.02 | 0.39±0.05 | 0.15±0.01 | 0.41±0.02 | 0.41±0.02 | 0.22±0.03 |
| ATISE | 0.35±0.01 | 0.49±0.03 | 0.15±0.01 | 0.47±0.01 | 0.40±0.00 | 0.21±0.01 |
| NE with $g'(\tau)$ | 0.99±0.00 | 0.98±0.00 | 0.90±0.00 | 0.64±0.00 | 0.66±0.00 | 0.32±0.00 |
| NE with $g(\tau)$ | 0.99±0.00 | 0.99±0.00 | 0.83±0.00 | 0.70±0.00 | 0.72±0.00 | 0.37±0.00 |

### B.3.2 Dimension of Embedding Vectors

Table 8 and 9 demonstrate that $d$ scarcely affects the performance of NE as long as it is more than some certain value. It is worth noting that NE's detection performance in GDELT may be further improved with larger values of $d$s, as illustrated in A.2.2, because $T_a = 2976$ in GDELT, which is much larger than the tested dimensions.

### B.3.3 Threshold for Valid TRs

In the main text, the threshold for distinguishing valid and invalid TRs is chosen as $\theta = 0.8$. Here we report NE's results on D14 with varying $\theta$s in Table 10. It is shown that NE still has an overwhelming advantage over all baselines.

### B.4 Mined TRs

Additional cases of mined TRs are shown in Table 11 as below.

Table 7: F1 of NE on ICEWS14 with different event type vectors

|    | Normalized | Unnormalized |
|----|-----------|-------------|
| F1 | 0.82      | 0.80        |

Table 8: NE on GDELT with different dimensions

| $d$ | 100 | 200 | 300 | 400 | 500 |
|-----|-----|-----|-----|-----|-----|
| TR Query (r) | 0.83 | 0.84 | 0.85 | 0.85 | 0.84 |
| TR Detection (F1) | 0.22 | 0.45 | 0.50 | 0.51 | 0.51 |

Table 9: NE on ICEWS14 with different dimensions

| $d$ | 50 | 100 | 200 | 400 | 600 | 800 |
|-----|----|----|----|----|----|----|
| TR Query (r) | 0.81 | 0.86 | 0.87 | 0.87 | 0.87 | 0.87 |
| TR Detection (F1) | 0.29 | 0.55 | 0.76 | 0.82 | 0.83 | 0.83 |

Table 10: Statistical results with different thresholds $\theta$ on ICEWS14

| Embedding | Detection(F1) | | | Query(r) | | |
|-----------|---------------|---------------|---------------|---------------|---------------|---------------|
|           | $\theta$=0.7 | $\theta$=0.8 | $\theta$=0.9 | $\theta$=0.7 | $\theta$=0.8 | $\theta$=0.9 |
| TNTComplEx | 0.29 | 0.26 | 0.25 | 0.07 | 0.08 | 0.06 |
| DE-SimplE | 0.28 | 0.22 | 0.21 | 0.09 | 0.09 | 0.21 |
| TASTER | 0.28 | 0.18 | 0.17 | 0.09 | 0.09 | 0.08 |
| TeRo | 0.35 | 0.43 | 0.40 | 0.08 | 0.08 | 0.06 |
| BoxTE | 0.38 | 0.40 | 0.41 | 0.08 | 0.18 | 0.06 |
| ATISE | 0.28 | 0.35 | 0.33 | 0.08 | 0.08 | 0.06 |
| NE with $g'(\tau)$ | 0.63 | 0.78 | 0.80 | 0.81 | 0.85 | 0.86 |
| NE with $g(\tau)$ | **0.71** | **0.82** | **0.84** | **0.87** | **0.87** | **0.87** |

Table 11: Cases of Mined TRs from ICEWS18

| Body Event | | | Head Event | | | $\tau$ | $gc$ | $sp$ |
|---|---|---|---|---|---|---|---|---|
| $s_b$ | $p_b$ | $o_b$ | $s_h$ | $p_h$ | $o_h$ | | | |
| North Korea | *Threaten with military force* | South Korea | North Korea | *Make empathetic comment* | South Korea | 87 | 0.86 | 3 |
| Bahrain | *Reduce or break diplomatic relations* | Foreign Affairs (United States) | Bahrain | *Consult* | Foreign Affairs (United States) | 0 | 0.86 | 3 |
| Japan | *Host a visit* | Yoshitaka Shindo | Yoshitaka Shindo | *Make a visit* | Japan | 0 | 1 | 7 |
| Government (Syria) | *Sign formal agreement* | Armed Rebel (Syria) | Armed Rebel (Syria) | *Sign formal agreement* | Government (Syria) | 0 | 1 | 5 |
| Protester (Thailand) | *Make statement* | Thailand | Protester (Thailand) | *Defy norms, law* | Military (Thailand) | 129 | 0.86 | 3 |
| China | *Make pessimistic comment* | Japan | China | *Host a visit* | Yasuo Fukuda | 148 | 0.83 | 5 |
| Court Judge (India) | *Express intent to cooperate* | Citizen (India) | Citizen (India) | *Accuse* | Villager (India) | 132 | 0.83 | 5 |
| Kim Jong-Un | *Appeal for diplomatic cooperation (such as policy support)* | South Korea | South Korea | *Engage in diplomatic cooperation* | Iran | 27 | 0.86 | 3 |
| South Korea | *Make pessimistic comment* | North Korea | Canada | *Sign formal agreement* | South Korea | 68 | 0.86 | 3 |
| China | *Appeal for diplomatic cooperation (such as policy support)* | Malaysia | South Korea | *Express intent to settle dispute* | China | 111 | 0.86 | 3 |
| Iraq | *Host a visit* | Massoud Barzani | Mohammad Javad Zarif | *Consult* | Massoud Barzani | 57 | 0.86 | 3 |
| Japan | *Appeal for diplomatic cooperation (such as policy support)* | Thailand | Citizen (Thailand) | *Fight with small arms and light weapons* | Thailand | 118 | 0.86 | 3 |

