# OpenReview forum: "Noether Embedding: Efficient Learning of Temporal Regularities"
_NeurIPS.cc/2023/Conference — NeurIPS 2023 poster_

### Official Review · Reviewer_7yn1 · 2023-06-27

**Soundness:** 2 fair
**Presentation:** 2 fair
**Contribution:** 2 fair
**Rating:** 5
**Confidence:** 2

**Summary:**

The paper presents a method for detecting, and embedding, temporal regularities from time-stamped event data, where events have a discrete type, and temporal regularities correspond to one base type preceding another head type by a characteristic relative time. A temporal regularity (TR) is defined in the following way. A temporal regularity associates a body event-type $b$ with a head-event type $h$, alongside a mean relative time $\tau$, and a universal factor $\eta$, such that events of type $b$ at time $t$ are "regularly" followed (or maybe also preceded) by an event of type $h$, with a mean time delay of $\tau$ but with a time variation in the interval $\Delta = [\tau(1-\eta), \tau(1+\eta)]$. One challenge of this is to discover these temporal regularities purely from time-stamped events, i.e. to abstract away from the specific time of the body event type. The principle contribution of this work is to train Noether Embeddings (NE) which the authors state "enable both the data efficient formation and rapid retrieval of TRs simply through embedding each event sample".


The authors define a collection of metrics for temporal regularities between any given body and head types (b,h). standard confidence (sc), head coverage (hc) and general confidence (gc), the last of which is a key measure of how strong the evidence is for a temporal regularity under a particular interval $\Delta$ (which is itself a consequence of the mean time $\tau$). They go on to specify two tasks, the first "TR detection" is to find the best value of general confidence for each pair of event types (for any value of relative-time) and specify this pair of event types as a TR if this value is above some threshold. The second task is TR Query which asks what the characteristic relative time between a pair of events is. Noether embeddings are then a composition of event embedding and time embedding, in such a way that the complex field is used to capture the time information withing just one part of the composition. TRs detection (or scoring) can then be efficiently calculated between any pair of embedded events. Associated methods are then used to determine the TR detection and query operations.

The authors compare their method of evaluating TR detection and querying with appropriately adapted methods from the literature. It isn't entirely clear from the paper, whether these other methods are trained to maximise scores between valid TRs in some way (but the assumption is that they are). The authors show that, on three datasets derived from large scale temporal knowledge graph data used elsewhere in the literature, their method performs substantially better at detecting and scoring (querying) temporal regularities in data.

**Strengths:**

The method is well motivated in terms of Emmy Noether's work corresponding symmetries with conservation laws. The authors develop a neat way to encode things that lends itself to efficient training and their performance on three different datasets is substantially better than other methods.
It isn't entirely clear how to judge the significance of this work, as the authors define their own task and their own metrics, possibly because there are no suitable pre-existing candidates for this.

**Weaknesses:**

The paper could be a little clearer in parts. For instance:
* It is unclear whether the authors' definition of temporal regularity (TR) is unique to the authors or is defined elsewhere. The authors relate this to prior work on temporal regularities, but the precise formulation appears to be the authors' own. It would be helpful to know whether and how this relates to other formal definitions of temporal regularities.
* More effort could be made to give intuitive (informal) descriptions of the defined terms, e.g. standard confidence, TR Detection, TR Query, ground truth confidence, and so on.
* How the data is partitioned into test and train, and how various methods are optimised from training data could be made clearer in the main body of the paper.
* It isn't really clear until late in the paper that the time-stamps are discrete. I am not sure whether this method would be tractable for continuously valued time-stamps
* The intuitive meaning of what is being established by temporal regularities and the associated metrics could be clearer too. For instance, what does it mean to have a ground-truth confidence greater than 0.8. Can that be given a (loose) statistical interpretation at all?
* The notion of an interval $\Delta$ which depends on the mean relative time $\tau$ is not clearly discussed. Is this like saying that the TRs model events whose relative time is uniformly distributed in some range? Is this a little brittle compared to a method that "smoothly" traded off relative-time-clustering with observation frequency?

The notation is a little confusing at times. In particular, Equation 2 or the text around it, could make some things clearer:
* that this metric applies to a specific pairing of event types b and h and a time interval (I realise this is wrapped up in the tr variable but I found the notation pretty opaque).
* that the tr, sp, hc, and gc terms depend on $\Delta$ but really more fundamentally depend on the mean relative time $\tau$. Later the authors refer to  $gc$ as a function of $\tau$ so this seems to me to be the fundamental variable. Would it be better to define tr as $(ev_b, ev_h, \tau, \eta)$ where $\Delta$ is derived from these values?
* that this metric is defined over an event set (by which I meant that the event set could appear formally in notation)
* b(q;tr) and n(b(tr)) mean two slightly different things, the latter is a count over b(q;tr) for all q in the event set I think. Similarly replacing b with h.

The definition of TR Detection and TR Query seem to be the authors own, and so the fairness of the experimental comparison is difficult to judge. The performance is impressive on first appearance, but I am left wondering how to interpret this. The paper would be greatly strengthened by a clear justification of why this is the fairest available comparison between methods.

**Questions:**

Why does the queried time set $\mathbb{T}_r = \{1-T_a,\ldots, 0, \ldots, T_a-1\}$ mean that you only query one side of an asymmetric pair of events? Surely this means that the total number of relative time durations is $2T_a -2$, meaning that any pair of events $(ev_i, ev_j)$ at times $t_i$ and $t_j$ would be queried in both directions so long as $t_i\neq 0$ or $t_j \neq T_a$ and vice versa.

What is the measure of TR Query (denoted r in table 1)?

How is the data in the experiments split into training and testing parts?

Why are there only 1 element per periodic time delay in the $\omega$ vectors? Does this mean that you can only discover 1 TR (body-head type) per characteristic time delay $\tau$? What if two distinct TRs (body-head types) had the same characteristic time delay?

**Limitations:**

The authors could be clearer about the limitations of this work. In particular, certain choices such as predefined fixed length interval factors for detection, predefined thresholds for detection, point estimates for TR Queries may be necessary to make the method tractable, but they do introduce a certain arbitrariness to the prediction frame work.
Equally, the fact that the experiments are comparing the performance of their method in optimising against an objective that motivated the design of their approach is potentially problematic too. The ground truth general confidence of a TR is the best performing time delay for a pair of events, which is the same as the objective for their detection method. The argmax of this is both the ground-truth for TR Query and the training objective for TR Query on their model. The authors should justify why this is a fair comparator, and possibly develop alternative independent measures for evaluation too.

---

> ### Author Rebuttal · Authors · 2023-08-09
>
> We sincerely appreciate the reviewer for spending valuable time reviewing our manuscript and providing insightful comments. We have improved our paper accordingly but discovered some misunderstandings concerning the content and contributions of the work. Our responses are provided below.
>
> **The foremost QA**
>
> **Q**: ‘The paper would be greatly strengthened by a clear justification of why this is the fairest available comparison between methods.’
>
> **A**: **The answer is in the second section of the global response at the top of the webpage.**
>
> **From ‘Weaknesses’**
>
> **Q1**: ‘whether TR is unique to the authors or is defined elsewhere’
>
> **A1**: Our definition is unique. A relevant definition for knowledge graph streams is in [1]. The main difference is that TR is suitable for arbitrary structured events, whereas the relevant definition in [1] applies only to knowledge graph streams. Additionally, TR introduces an adaptive $\eta$ to allow vibrations of $\tau$ in real-world data distributions.
>
> **Q2**: ‘intuitive (informal) descriptions of the defined terms…’
>
> **A2**: We have revised our paper accordingly. For example, the intuitive description of standard confidence can be viewed as the probability that the head event will occur at time $t+\bigtriangleup$ once the body event occurs at time $t$.
>
> **Q3**: ‘How the data is partitioned into test and train, and how various methods are optimised from training data…’
>
> **A3**: The train and test sets are of the same set. To enhance understanding, we can draw a comparison between TR learning and clustering. In this analogy, event items correspond to clustered samples, and TRs correspond to the clusters. Therefore, NE can be viewed as a memory with unsupervised learning capabilities due to its specific structural biases.
>
> The loss functions of NE and existing embeddings all separate the score functions of positive and negative event samples, treating representation learning as a two-class classification problem. All the baseline embedding models adopt the log-softmax form of loss functions, as in their original settings.
>
> **Q4**: ‘whether this method would be tractable for continuously valued time-stamps’
>
> **A4**: It would be tractable because continuously valued time stamps can be discretized.
>
> **Q5**: ‘what does it mean to have a ground-truth confidence greater than 0.8.’
>
> **A5**: This implies that there is an 80% probability for the co-occurrence of a body event at time $t$ and a head event at time $t+\bigtriangleup$.
>
> **Q6**: ‘The notion of an interval $\bigtriangleup$ which depends on the mean relative time $\tau$ is not clearly discussed’
>
> **A6**: Since such a definition is used for calculating metrics such as general confidence, it is required to be independent of specific data distributions. Intuitive meaning: the tolerance of noises (width of $\bigtriangleup$) is proportional to the size of relative time $\tau$.
>
> **Q7**: ‘The notation is a little confusing at times. In particular, Equation 2 or the text around it, could make some things clearer’
>
> **A7**: We have made improvements in the revised paper. For example, we have changed $tr: (ev _b,ev _h,\bigtriangleup)$ into $tr: (ev _b,ev _h,\tau,\eta)$ as suggested.
>
> **From ‘Questions’**
>
> **Q1**: ‘Why does the queried time set mean that you only query one side of an asymmetric pair of events?’
>
> **A1**: Considering two event types $ev _i, ev _j$. Since $g(\tau;ev _i,ev _j)=g(-\tau;ev _j,ev _i)$, it is only necessary to calculate one decoding function between  $g(\tau;ev _i,ev _j)$ and $g(-\tau;ev _j,ev _i)$ at the decoding stage.
>
> **Q2**: ‘What is the measure of TR Query’
>
> **A2**: As stated in Section 2.2, TR query is to output the correct $\tau'=\tau _g$ for valid TRs, where the ground truth $\tau_g$ is set as what maximizes gc(\tau) in computing $gc_g$. For each tested query $(ev _b,ev _h)$, a ratio $r'=\frac{gc(\tau')}{gc _g}$ is computed. The averaged ratio $r$ of all queries is reported.
>
> **Q3**: ‘…training and testing parts?’
>
> **A3**: The same as in Answer 3 from ‘Weaknesses’.
>
> **Q4**: ‘Why are there only 1 element per periodic time delay in the $\omega$ vectors?’ & ‘What if two distinct TRs (body-head types) had the same characteristic time delay?’
>
> **A4**: One advantage of NE is exactly its ability to efficiently store large amounts of intertwined TRs. The specific reason is in 3(2) of the global response at the top of the webpage.
>
> **From ‘Limitations’**
>
> **Q1**: ‘they do introduce a certain arbitrariness to the prediction frame work’
>
> **A1**: We have conducted data analysis to rationalize the parameters as much as possible. For example, we have analyzed that TRs whose gc ∼ 0.8 constitute a small percentage of all tested TRs. We have also made reasonable adjustments. For example, we have conducted experiments in Appendix C.3.3 where the threshold for distinguishing valid and invalid TRs is chosen as 0.7,0.8,0.9, respectively. We emphasize that, similar to the classic clustering task, some degree of ‘arbitrariness’ is inevitable due to the unsupervised nature of TR learning.
>
> **Q2**: ‘justify why this is a fair comparator ... independent measures for evaluation too’
>
> **A2**: Fairness is ensured by treating both NE and existing embeddings equally as justified in the global response. Moreover, we disagree with the statement that ‘the argmax is … for TR Query on their model’. At the decoding stage for TR query, the argmax in evaluation is performed on the relative time. At the training stage, however, no information about the relative time is needed. The training loss is only relevant to the specific information $(ev, t)$ of each event sample and whether it is a positive or negative sample.
>
> We have greatly improved the paper by the reviewer's valuable feedback. We emphasize that NE's advantage of 'efficiency' is evident even without the comparative experiments.
>
> **Reference**
>
> [1] Omran P G, Wang K, Wang Z. Learning Temporal Rules from Knowledge Graph Streams. AAAI, 2019.

---

> > ### Comment · Area_Chair_W4BB · 2023-08-18
> >
> > Hi Reviewer,
> >
> > This paper has divergent scores. So, please give your feedback after reading author rebuttal and other reviewers' comments.
> >
> > Your AC

---

> > ### Comment · Reviewer_7yn1 · 2023-08-21
> >
> > Thank you for your detailed responses to my questions.
> >
> > I note your preference to refocus the paper "as a first ‘efficient’ TR learner with event embeddings" and I think this is a good choice. I still think there are some weaknesses to the paper and much of this aligns with my original review. In particular I think it would help to define temporal regularity formally, independently of your detection mechanism. In the paper, you write "We define the simplest and most basic 1-1 TR...". This implies that you at least have an intuitive notion of what a more general description would be. And that is before we challenge the use of 1-1 (what would be a non 1-1 TR?).
> >
> > Something informed by causal theory would probably be useful here (e.g. see the Pearl textbook or one of the many more recent works in the field). You could also relate TRs to other formalisms for reasoning about temporal events, such as "the event calculus" or "temporal relations". Most valuable would be a clear articulation of what you think a TR actually is, and then define some subclass (or set of nested subclasses) that indicate the simplifying assumptions you are making).
> >
> > For instance, to my understanding, causal theory states that correlations between two random variables, X and Y, can arise if one causally influences the other directly, or if there is some third variable (or set of variables) that influence both (or other more complex relationships). The fact that your TR formalism is sensitive to order, implies that it would be more sensitive to the former case than the latter (i.e. one RV causally influencing another). In that case, one could still define a model that makes weaker assumptions than you do. For instance, that the causal RV (let's say X) induces a distribution over the wait time before which Y is observed. Your formalism (as I indicated in my original review) makes further assumptions that a) the distribution is uniform in some interval b) that the interval width is determined by the mean wait time.
> >
> > This is, I think, highly relevant for work that claims to be the first meaningful attempt to detect TRs.
> >
> > I also have some issues with some of your descriptions about the meaning of metrics. In particular, in response to my question:
> > >Q5: ‘what does it mean to have a ground-truth confidence greater than 0.8.’
> >
> > You reply:
> > > A5: This implies that there is an 80% probability for the co-occurrence of a body event at time $t$ and a head event at time $t+\Delta$.
> >
> > But I think that your descriptions elsewhere are more accurate, namely that it is the probability that you will see a head event at time $t+\Delta$ given that you have observed a suitable body event at time $t$.
> >
> > Finally, I think that my original statement about your experimental conditions limiting the possibility of recognising two distinct TRs with the same mean wait time (something that your general formalism doesn't restrict).
> >
> > Nonetheless, I have read the other responses and reviews, and I feel that I can be persuaded to increase my rating to weak accept based on three main strengths of the paper:
> > * The novelty of the formalism and its relationship to Noether's theorem. The mathematical structure, which utilises complex numbers to capture characteristic wait times, may well have a much wider applicability and I think raises the interest of this paper significantly.
> > * The experiments are well constructed and the results convincing.
> > * The fact that there are no existing methods (that I know of) that can detect temporal regularities with this kind of efficiency.

---

> > > ### Author Response · Authors · 2023-08-21
> > > **New Response**
> > >
> > > **Thank you for your response.**
> > >
> > > 1.	We greatly appreciate your suggestion to provide a clear articulation of what a TR actually is, 'independently of our detection mechanism'. We will incorporate your suggestion in the revised paper to enhance clarity.
> > >
> > > 2.	Regarding Q5, we understand that your interpretation of ‘ground-truth confidence’ aligns with the ‘global confidence’ in our paper. Our response is appropriate if our understanding is correct (inappropriate if wrong). In that context, your preferred answer corresponds to the ‘standard confidence’ in the paper, which is combined with the 'head coverage' to form the ‘global confidence’ using a harmonic mean.
> > >
> > > 3.	In our comparative experiment setting, we excluded the scenario where $\tau=0$ because it represents a large proportion of all TRs but does not effectively reflect the difficulty of the TR query task. Baselines can easily handle queries with $\tau=0$ as they are too simple. In our grouped experiment (Figure 4(c)), we specifically test this scenario to demonstrate the flexibility of NE.
> > >
> > > We appreciate your increase in score and thanks again for your further response and precious suggestions. **Please note that ‘weak accept’, as you suggested, corresponds to a score of 6 rather than 5 (denoting borderline accept), as you previously assigned**.

---

> > > > ### Comment · Reviewer_7yn1 · 2023-08-21
> > > >
> > > > I intended to say Borderline Accept. My score is 5.
> > > >
> > > > On 2., I still do not know what you mean by:
> > > > > This implies that there is an 80% probability for the co-occurrence of a body event at time $t$ and a head event at time $t+\Delta$.
> > > >
> > > > The paper states that $gc(tr)$ is the *general confidence* of a temporal regularity $tr=(ev_b, ev_h, \Delta)$, or possibly more readably $tr=(ev_b, ev_h, \tau, \eta)$ (as discussed). This is the harmonic mean between the standard confidence and head coverage. The standard confidence is a bit like an estimation of $p(ev_h @ t+\Delta | ev_b @ t)$ - in words, the probability that an event of type $ev_h$ occurs within the specified interval after an event of type $ev_b$ observed previously.
> > > > The head coverage is a bit like an estimation of  $p(ev_b @ t - \Delta | ev_h @ t )$ - in words, the probability that an event of type $ev_b$ occurs within the specified interval before an event of type $ev_h$ observed at some later time. However, you appear to be claiming that the average of these two values is, in some way, equivalent to some joint probability.
> > > >
> > > > Put another way, let's consider an extreme case where you have 100 time-points, and six events. Three events, $q_a=(ev_b,t)$, $q_b=(ev_b,t')$, $q_c=(ev_b,t'')$ and three events $q_d=(ev_h,t+\tau)$, $q_e=(ev_h,t'+\tau)$ and $q_f = (ev_h,t''+2*\tau)$ and these are the only events in your system. Consider the temporal regularity $tr=(ev_b, ev_g, \tau, \et=0.2)$ then:
> > > > \begin{align}
> > > > n(b(tr)) & = n(h(tr)) = 3
> > > > \\
> > > > sp(tr) = 2
> > > > \\
> > > > sc(tr)=hc(tr)= 2/3=gc(tr)=gc_g
> > > > \end{align}
> > > > In other words, the general confidence is exactly $0.67$. But you claim that you would then interpret that as:
> > > >
> > > > > This implies that there is a 66.7% probability for the co-occurrence of a body event (ev_b) at time $t$ and a head event (ev_h) at time $t+\Delta$.
> > > >
> > > > Putting aside concerns about over-precise estimates based on frequentist approximations, what does that actually mean? It seems to refer to something like a confidence that the observed temporal regularity is a meaningful pattern, but this is hard to turn into a probability. Consider instead that you observed many more such events, e.g. $n(b(tr))=n(h(tr))=45$ such that $sp(tr)=30$ then you would again have a 66.7% probability for this co-occurence but under this second scenario you would be much more certain that a pattern exists. Your statement seems to be over-claiming here.
> > > >
> > > > On 3. I say:
> > > > > Finally, I think that my original statement about your experimental conditions limiting the possibility of recognising two distinct TRs with the same mean wait time (something that your general formalism doesn't restrict).
> > > >
> > > > Which you interpret as $\tau=0$, but that isn't what I mean. I mean that you have two temporal regularities $(ev_b, ev_h, \tau, \eta)$ and  $(ev'_b, ev'_h, \tau', \eta)$ where $\tau=\tau'>0$ but $ev_b \neq ev'_b$ and $ev_h \neq ev'_h$. The way you construct your $\omega$ vector in the experiments appears to me to suggest that there is some element of $\omega$, let's call it $\omega_k$ that is associated with temporal regularities with a wait time of $\tau$, and therefore those two distinct TRs would clash. Is that an accurate interpretation?

---

> > > > > ### Author Response · Authors · 2023-08-21
> > > > > **Further response**
> > > > >
> > > > > Thanks for your detailed analysis.
> > > > >
> > > > > 1.	We agree that probability is just an approximate explanation because support $sp$ is also needed to ensure statistical sufficiency. We have clarified this in the revised paper. We sincerely consult the reviewer if a better explanation can be provided. We emphasize that the concepts of ‘standard confidence’, ‘head coverage’, and ‘general confidence’ are commonly used in the rule mining field, and we borrow evaluation metrics from it mainly to ensure fair and reasonable evaluations.
> > > > >
> > > > > 2.	They will not clash as explained below. Since $g(\tau;ev _b,ev _h)=2-2\sum _{j=1}^d Real(\overline{\pmb{u} _b} \circ \pmb{u} _h \circ e^{i \pmb{\omega} \tau}) _j\in [0,4]$, if $g(\tau;ev _b,ev _h)=g(\tau’;ev’ _b,ev’ _h)=0$ (implying perfect TRs), we have $\sum _{j=1}^d Real(\overline{\pmb{u} _b} \circ \pmb{u} _h \circ e^{i \pmb{\omega} \tau}) _j = \sum _{j=1}^d Real(\overline{\pmb{u} _b’} \circ \pmb{u} _h’ \circ e^{i \pmb{\omega} \tau’}) _j =1$. Since $||\pmb{u}||=1$, we have $(\overline{\pmb{u} _b} \circ \pmb{u} _h \circ e^{i \pmb{\omega} \tau}) _j=(\overline{\pmb{u} _b’} \circ \pmb{u} _h’ \circ e^{i \pmb{\omega} \tau’}) _j=\frac{1}{d}, j=1,…,d$. This exactly means that $\overline{\pmb{u} _b}$ is conjugate with $\pmb{u} _h \circ e^{i \pmb{\omega} \tau}$ in all $d$ complex dimensions (the same relationship for $\overline{\pmb{u} _b’}$ and $\pmb{u} _h’ \circ e^{i \pmb{\omega} \tau’}$).

---

> > > ### Author Response · Authors · 2023-08-22
> > >
> > > **On the definition of TR, we have made improvements in the revised paper by the reviewer's comment. We sincerely ask the reviewer if the following description is acceptable:**
> > >
> > > By temporal regularity we refer to the building structure of event schemas [1]. According to cognitive science literature, event schemas are learned directly from experience by accumulating common event structures statistically. These schemas also possess chronological relationships [2]. Building on the statistical interpretation, we formally define temporal regularity as temporal associations that remain invariant to time shifts: $ ( ev_b, t ) \to ( ev_h, t + \tau) \quad \forall t \in T_a $. Here, $\tau=0$ represents the synchrony of event occurrences, and both $t$ and $\tau$ can be either discrete or continuous. Since real-world data distributions often contain noise, we introduce an adaptive $\bigtriangleup=[\tau(1-\eta), \tau(1+\eta)]$ to replace $\tau$ when evaluating statistically learned temporal regularities from events. This evaluation approach offers the advantages of both simplicity and practicality by allowing for intuitive vibrations.
> > >
> > > **We appreciate again the reviewer for your constructive comments.**
> > >
> > > References
> > >
> > > [1] Pudhiyidath A, Roome H E, Coughlin C, et al. Developmental differences in temporal schema acquisition impact reasoning decisions[J]. Cognitive Neuropsychology, 2020.
> > >
> > > [2] Ghosh V E, Gilboa A. What is a memory schema? A historical perspective on current neuroscience literature[J]. Neuropsychologia, 2014.

---

> ### Author Response · Authors · 2023-08-21
> **Willingness to answer further questions**
>
> Dear reviewer 7yn1
>
> We thank you for your precious time and constructive comments. As the discussion period will end soon, we are not sure whether our responses have addressed your questions. If you still have any questions about our work, we are more than happy to provide further responses for you.

---

### Official Review · Reviewer_zRYw · 2023-07-07

**Soundness:** 2 fair
**Presentation:** 2 fair
**Contribution:** 2 fair
**Rating:** 4
**Confidence:** 3

**Summary:**

This paper defines the tasks of temporal regularity (TR) detection and query and their evaluation metrics, and proposes Noether Embedding (NE) that enables encoding TRs from limited event samples and rapid retrieval of TRs in constant time complexity. NE possesses time-translation symmetries of temporal regularities that are indicated by conserved local energies in the embedding space. NE is evaluated on ICEWS14, ICEWS18, and GDELT datasets, and achieves about double F1 scores for detecting valid TRs and over 10 times confidence scores for querying TR intervals compared with baseline embeddings with additional calculation. NE is further shown to be useful for social event prediction and personal decision-making scenarios.

**Strengths:**

- This work defines a pair of particularly novel tasks: temporal regularity detection and query. Both are critical to human-level intelligence.
- It proposes Noether Embedding which for the first time enables learning temporal regularities directly from events and rapid retrieval of the regularities.
- NE achieves promising performances on ICEWS14, ICEWS18, and GDELT datasets, and is shown to be useful for social event prediction and personal decision-making scenarios.

**Weaknesses:**

Please see Questions for doubts to be addressed in the next version of the paper.

**Questions:**

1. L279 argues that NE is qualitatively different from rule mining methods in that a search method may require fixing different relative time points before mining rules, while NE enables direct fitting and plotting of an approximate validity distribution of relative time points. But L150-151 suggests that users will select a set of relative time points. So how does NE differ? If users provide their selected set of relative time points, why couldn't we detect time regularities by counting?
2. What structural biases do NE assume? Could you provide theoretical analyses for the capacity limitation of NE?
3. Could you add to Section 3.2 the geometric interpretation of the Hardmard product in Eq. (4)?
4. L116: Is the $\omega$ vector a hyperparameter or learned, and what is the semantic implication of it being certain values?
5. L122: Should $t^\prime$ exclude all $\hat{t}$ where $(ev,\hat{t})$ is in the dataset?
6. To confirm understanding, the second equal sign is L134 always hold true, and by construction, g being invariant to t is always true. Right?

**Limitations:**

The Conclusion section mentions limitations. The paper doesn't seem to discuss negative societal impacts. It could say something about when the method may fail to detect or query temporal regularities in applications and potential consequences.

---

> ### Author Rebuttal · Authors · 2023-08-09
>
> We sincerely appreciate the reviewer for spending valuable time reviewing our manuscript and providing insightful comments. We have improved our paper accordingly, and our responses are as below.
>
> **Q1**: ‘So how does NE differ?’ & ‘why couldn't we detect time regularities by counting?’
>
> **A1**: The main difference lies in the transfer of time complexity. Specifically, the counting time in the search for each $(ev _b,ev _h,\tau)$ is proportional to the number of relevant events. This time complexity is transferred to the training stage of NE so that decoding each $(ev _b,ev _h,\tau)$ is only O(d), and even O(1) by applying GPUs. Therefore, the NE vectors after the training stage are functionally equivalent to an approximate memory of all TR validity results after the counting process for each $(ev _b,ev _h,\tau)$. We have made this clearer in the revised paper.
>
> **Q2**: ‘What structural biases do NE assume?’
>
> **A2**: NE’s structural biases are directly inspired by Noether’s theorem. Specifically, (1) the event embedding $\pmb{q}(t;ev)$ should be constructed to make each local energy $g$ invariant to $t$; (2) the training loss should be constructed to make the value of $g$ approximate TR validity; (3) we should use local energies $g$s as the decoding function. The detailed reasons for how these biases enable NE are explained in Section 3.3 and theoretically analyzed in Appendix B.2.
>
> **Q3**: ‘Could you provide theoretical analyses for the capacity limitation of NE?’
>
> **A3**: In Appendix B.2.2, we theoretically analyzed two inequality constraints that control NE’s performance. We discovered that the vector dimension $d$ is better to be larger than the number of relative time points $T _a$ to enable NE. Such a requirement is also confirmed by the ablation experiment in Table 5 in Appendix C.3.2. We will further provide a more strict proof of this capacity limitation in the next version of our paper.
>
> **Q4**: ‘Could you add to Section 3.2 the geometric interpretation of the Hardmard product in Eq. (4)?’
>
> **A4**: The Hardmard product can be depicted as a rotation of event-type vectors by time in the d-dimensional complex space. We have added this interpretation in our revised paper.
>
> **Q5**: ‘Is the $\omega$ vector a hyperparameter or learned, and what is the semantic implication of it being certain values?’
>
> **A5**: The $\pmb{\omega}$ vector is manually set as an exponential distribution in the paper. We also show in Section 4.4 (Ablation Studies -- Frequency Distribution) that the exponential distribution surpasses linear distribution for fitting larger datasets.
>
> $\pmb{\omega}$ provides the basis for Fourier-like expansions. Specifically, since the score function $f(t;ev)=\sum _{j=1}^d Real(\pmb{u} \circ e^{i \pmb{\omega} t}) _j$ and the decoding function $g(\tau;ev _b,ev _h)=2-2\sum _{j=1}^d Real(\overline{\pmb{u} _b} \circ \pmb{u} _h \circ e^{i \pmb{\omega} \tau}) _j\in [0,4]$ can be viewed as Fourier-like expansions, we can see that the global time vector $\pmb{\omega}$ provides the expansion basis and the event-type vectors $\pmb{u}$s store the coefficients for $f(t)$ (revealing event occurrence) and composing $\overline{\pmb{u} _b} \circ \pmb{u} _h$ as the coefficients for $g(\tau)$ (revealing TR validity).
>
> **Q6**: ‘Should $t’$ exclude all $t$ where (ev,t) is in the dataset?’
>
> **A6**: Your comment is reasonable. Exclusion is needed in the paper for rigor’s sake. However, it is also acceptable if it does not, as it trades a little performance drop for a faster implementation. This is because negative samples contribute much less ($\frac{1}{N}$) than a positive sample ($1$) as long as the number of negative samples $N$ is much larger than $1$, which is the general case. Therefore, those positive samples ‘wrongly regarded’ as negative samples will make a minor difference in the loss score.
>
> **Q7**: ‘the second equal sign is L134 always hold true, and by construction, $g$ being invariant to $t$ is always true. Right?’
>
> **A7**: Yes, it does. This contributes to NE’s major structural bias as discussed in Answer 2.
>
> **Q8**: ‘The paper doesn't seem to discuss negative societal impacts’
>
> **A8**: NE fails if cannot fit the whole dataset well, which is generally when the vector dimension $d$ is set smaller than the number of absolute time points $T _a$. The potential consequence is that false positives may lead to inaccurate prediction and that false negatives may lead to untimely warnings. We have added this discussion to the revised paper.
>
> Once again, we would like to express our gratitude to the reviewer for the valuable feedback, which has helped us further improve our manuscript.

---

> > ### Comment · Reviewer_zRYw · 2023-08-21
> >
> > Dear Authors,
> >
> > Many thanks for your reply, which addressed multiple of my doubts. I still have a major concern about the usefulness of the embedding, as compared to counting. To know if there is a time regularity between two event types, one could, for example, check their pairwise cooccurrences and see if the time offsets form a unimodal distribution. This simple counting method can detect time-invariant time regularities.
> >
> > You indicated that the time-complexity of counting is transferred to the embedding training stage; but counting can also happen offline at "training" time. You argued that one of the two main advantages of NE is data-efficient learning; but counting can always use the same amount of data to obtain the same or better detection accuracy. Counting should be the default choice since it avoids unnecessary structural biases of using certain score and decoding functions. While your proposed functions satisfy the desiderata of time-invariance, they introduce additional priors that are hard to interpret.
> >
> > In terms of efficient querying, a simple lookup dictionary obtained by counting will be O(1) to query. Alternatively this lookup dictionary can be compressed in various ways to consume less space.
> >
> > Could you help me understand the scenario when (each component of) NE is more favorable than counting? In any case, counting methods seem to be necessary baselines in the paper. These are my current doubts, and I hope to learn thoughts from authors and other reviewers. Meanwhile, I have carefully read the comments from other reviewers and agree that clarification and revision are needed before the work is more complete.
> >
> > Nevertheless, I acknowledge the importance of the proposed tasks and appreciate the existing experiments and insights, which provide much food for thought. I hope the next version of the paper will address the issues raised in the rebuttal conversations.

---

> > > ### Author Response · Authors · 2023-08-21
> > > **The senario does exist**
> > >
> > > Thank you for your additional question. It is worth noting that there is a specific scenario in which NE functionally surpasses counting, and that is when vectors are required for storage and querying purposes. The emergence of large language models has led to a rapid growth in vector databases, with many startups entering this domain recently. Looking ahead, there is a high likelihood that vector databases will become the prevailing storage solution for various data types, encompassing unstructured data, structured knowledge, structured events, and more. Given this future trend towards vector storage dominance, our research holds significant potential utility.

---

> > > > ### Comment · Reviewer_zRYw · 2023-08-21
> > > >
> > > > Thank you. I don't have enough knowledge about vector databases. Abstractly I understand vector representations are popular and there will be software and hardware designed for vectors. Concretely, I'm not sure when it's necessary to use those vector databases. Even if we want to use them, I suspect there will be a method to convert the "counted" TRs into vector representations which can be tractably queried and are more accurate than vectors learned using NE, due to the uninterpretable bias from using certain score and energy functions.
> > > >
> > > > I think including counting as a baseline is needed: counting should be more accurate at TR detection and actually provide the ground truth, while NE may or may not be a more compact way of (imperfectly) storing the same TRs; both can be efficiently queried. It may be the case that counting is intractable in some sense, but then approximated counting methods can be considered as baselines. It could also happen that NE is similarly accurate than vector representations obtained from counted TRs, but it'd be nice to see what happens empirically. As of now I don't see sufficient evidence that NE for TR detection is favored or necessary and NE for TR querying is more efficient than simple lookup.
> > > >
> > > > Nevertheless I appreciate the proposed tasks and am eager to hear more comments.

---

> > > > > ### Author Response · Authors · 2023-08-21
> > > > > **A near scenario**
> > > > >
> > > > > A near scenario is when the occurrences of different event types are stored separately. **One uniqueness of NE (compared to counting) is that the training unit is only an event sample, rather than TRs. Consequently, the event-type vectors of different event types can be trained separately, utilizing different GPUs and stored in different vector databases each with no information visible to each other.** These vectors can then be quickly combined when required to efficiently query TRs. By ensuring that a common protocol sets the same global time vector, events in the world can be encoded separately by training different event-type vectors. NE then enables the efficient detection and querying of TRs when required. **Counting, instead, cannot handle this situation.**

---

> > > > > ### Author Response · Authors · 2023-08-21
> > > > > **Further Response**
> > > > >
> > > > > **Thank you for inspiring us to explore the potential use of NE from an application perspective.** We have incorporated this discussion into our revised paper. Thanks for your suggestion about the counting baseline. However, **we prefer comparing NE to counting by text description because the uniqueness of NE, as explained in the former response, does not require experimental evidence.** Besides, our main objective is to advance the representation learning field. **It is worth noting that the efficient formation of structures using limited data has traditionally been a challenge for distributed representations.** Typically, deep learning methods rely on a substantial amount of lower-level data to learn higher-level structures, such as concepts learned in CNNs from images [1] and grammars learned in GPT from sentences [2]. In our study, we focus on learning higher-level TRs in event embeddings with only limited lower-level events (depicted in Figure 4). **Therefore, the problem we tackle is not only crucial but also highly challenging.**
> > > > >
> > > > > References
> > > > >
> > > > > [1] Chen, Zhi, Yijie Bei, and Cynthia Rudin. "Concept whitening for interpretable image recognition." Nature Machine Intelligence, 2020.
> > > > >
> > > > > [2] Mahowald, Kyle, et al. "Dissociating language and thought in large language models: a cognitive perspective." arXiv, 2023.

---

> > > > > > ### Comment · Reviewer_zRYw · 2023-08-21
> > > > > >
> > > > > > Thank you for the prompt response.
> > > > > >
> > > > > > I will instead claim that the common purpose of designing structure priors is exactly data efficiency. If we already know structural properties of the data, we let model architectures reflect them so that the embedding or network doesn't need to discover them from many data and relying on good learning algorithms. "Efficient formation of structures using limited data" is a common theme that the community is addressing and nothing unique.
> > > > > >
> > > > > > (1) Using CNN as an example, it is impossible to "count" all the pixel combinations. Observing the spatial-invariance property, CNN proposes to process image representations in a sliding window fashion so that computation becomes more tractable and less data are needed to learn local patterns, where "local" can mean different receptive field sizes and ultimately become "global". Arguably people may as well call CNNs "Noether networks". (2) For language, it is impossible to "count" all token combinations. The attention mechanism compares tokens in pairs and then aggregate information at each token position. This is apparently more data efficient than assuming no sequence structure and do a fully connected layer on the concatenation of all token features. (3) In the recent literature of relatively shallow embeddings including those you cited, people address both expressivity and inductive biases of models. However, the expressivity analysis is missing in your presentation of NE.
> > > > > >
> > > > > > The 1-1 TR tasks are much simpler than image or language tasks, or graph tasks that people recently propose embedding methods for, or the classical rule mining tasks where counting algorithms are extensively explored. In 1-1 TR, counting is possible in polynomial time and tractable, but you don't seem to have carefully investigated or be interested in its advantages and disadvantages compared to NE. Counting should be the ground truth method but possibly have space disadvantage. You mentioned training speed advantage in the "near scenario", but counting can also be parallelized on many CPUs. I described a simple counting method in a previous message, and am not sure which part of the near scenario can counting not handle. Further, is there an accuracy guarantee of NE since it learns event type embeddings separately and uses certain score and energy functions that inevitably introduce uninterpretable biases? To achieve the same accuracy, how does the space complexity of NE and counting compare? I've also mentioned the possibility of converting counted TRs into vector representations instead of using NE for less accurate TR detection. The claims in your recent comments, both about general understanding of representation learning and about NE vs. baselines, may require additional supporting evidence.
> > > > > >
> > > > > > Nevertheless, I appreciate the fruitful conversation and look forward to the paper's improvement. Hope you will also correct me if I misunderstood or overlooked any important details.

---

> > > > > > > ### Author Response · Authors · 2023-08-21
> > > > > > > **Response 1**
> > > > > > >
> > > > > > > Thank you for your detailed response.
> > > > > > >
> > > > > > > 1.	We do adhere to the common theme addressed by the community, which involves incorporating structural priors into model architectures for more data-efficient learning. Our unique contribution lies in developing the first event embedding with specifically-designed structural priors that enable efficient learning of TRs, surpassing the capabilities of existing embeddings.
> > > > > > >
> > > > > > > 2.	When we mention that ‘the efficient formation of structures using limited data has traditionally been a challenge for distributed representations’, we are referring to achieving human-level data-efficient capabilities. For instance, probabilistic program induction outperforms CNN in achieving human-level data-efficient concept learning [1], while CNN (which employs distributed representations) requires significantly more data compared to humans. Hence, distributed representations have historically struggled to attain human-level data efficiency. However, our development of NE allows embeddings to exhibit human-level data-efficient learning capabilities for TRs [2][3][4], as depicted in Figure 4. This outcome defies common sense, even though we align with the common theme.
> > > > > > >
> > > > > > > 3.	As for ‘expressivity analysis’, since NE can be perceived as memory, its important advantage is that it enables the incorporation of both strong biases and large-capacity storage simultaneously. This aspect poses a genuine challenge in our model design. Appendix B.2 presents an analogical capacity analysis (in contrast to an expressive analysis), demonstrating that the vector dimension $d$ needs to exceed the number of absolute time points $T_a$ for good fitting and thus enable the model. We will further improve the analysis in our revised paper.
> > > > > > >
> > > > > > > Reference
> > > > > > >
> > > > > > > [1] Lake B M, Salakhutdinov R, Tenenbaum J B. Human-level concept learning through probabilistic program induction[J]. Science, 2015.
> > > > > > >
> > > > > > > [2] Hudson J A, Fivush R, Kuebli J. Scripts and episodes: The development of event memory[J]. Applied Cognitive Psychology, 1992.
> > > > > > >
> > > > > > > [3] Bauer P J, Mandler J M. One thing follows another: Effects of temporal structure on 1-to 2-year-olds' recall of events[J]. Developmental psychology, 1989.
> > > > > > >
> > > > > > > [4] Schapiro A C, Turk-Browne N B, Botvinick M M, et al. Complementary learning systems within the hippocampus: a neural network modelling approach to reconciling episodic memory with statistical learning[J]. Philosophical Transactions of the Royal Society B: Biological Sciences, 2017.

---

> > > > > > > ### Author Response · Authors · 2023-08-21
> > > > > > > **Response 2**
> > > > > > >
> > > > > > > We want to emphasize that **our main problem** is how to enable event embeddings with an efficient TR learning capability, which is distinct from those in the temporal rule mining field (generally using counting). Specifically, temporal rule mining is typically studied for practical applications, aiming to uncover event regularities in specific domains [1] [2] [3]. Our goal, instead, is to **advance the representation learning field** by enabling embedding models to efficiently learn the atomic structure (TR) of event schemas, **similar to how humans learn** [4][5].
> > > > > > >
> > > > > > > **We appreciate again for the reviewer's valuable questions.**
> > > > > > >
> > > > > > > Reference
> > > > > > >
> > > > > > > [1] Segura‐Delgado A, Gacto M J, Alcalá R, et al. Temporal association rule mining: An overview considering the time variable as an integral or implied component. DMTD, 2020.
> > > > > > >
> > > > > > > [2] Chen M, Chen S C, Shyu M L. Hierarchical temporal association mining for video event detection in video databases. IEEE, 2007.
> > > > > > >
> > > > > > > [3] Yoo J S, Shekhar S. Similarity-profiled temporal association mining. IEEE, 2008.
> > > > > > >
> > > > > > > [4] Pudhiyidath A, Roome H E, Coughlin C, et al. Developmental differences in temporal schema acquisition impact reasoning decisions. Cognitive Neuropsychology, 2020.
> > > > > > >
> > > > > > > [5] Chaudhuri R, Fiete I. Computational principles of memory. Nature neuroscience, 2016.

---

> > > > > > > > ### Comment · Reviewer_zRYw · 2023-08-21
> > > > > > > >
> > > > > > > > Dear Authors, many thanks for your prompt reply. Given your argument that NE has strong biases and large-capacity storage, I feel confident that its advantages can be proved for certain resource constraint scenarios in the next version of the paper. Nevertheless, as mentioned, counting should be the most accurate method given any amount of data. I don't have evidence that any other method with the same amount of data can be more accurate than counting. I am also not sure if it's meaningful to talk about human-level data efficiency. Humans have been processing innumerable data since we are born and much knowledge is transferrable. It remains mysterious how human brains work but it's clear that brain mechanisms are largely different from these task-specific models we learn. I agree that advancing representation learning is important, but since there's no previous baseline presentation learning methods, your pioneering work needs to show how embeddings are useful in this case at all. As mentioned, when we use CNN to process images or transformers to process sequences, it's clear that naive counting is intractable. But in your case, the motivation of using embeddings needs to be shown for the exact scenarios, now that counting should be tractable (polynomial time). I find the tasks intriguing and look forward to evidence that NE has better accuracy-efficiency trade-off compared to baselines with the minimal inductive biases and similar inductive biases in certain cases.

---

> > > > > > > > > ### Author Response · Authors · 2023-08-21
> > > > > > > > >
> > > > > > > > > Considering another scenario where N event types have already been trained. When the event occurrences of a new event type is suddenly known, with NE you only need to train the new event type, and then can you easily detect and query TRs between the new event type and the existing N event types. Counting can not naturally handle this scenario, either.

---

> > > > > > > > > ### Author Response · Authors · 2023-08-21
> > > > > > > > >
> > > > > > > > > We sincerely appreciate the reviewer for recognizing the pioneering aspect of our research. We also agree with the reviewer that NE may have wide potential use. Since the content of this paper can already support its motivation to advance the representation learning field, we kindly request the reviewer to make a re-evaluation.

---

> > > > > > > > > > ### Comment · Reviewer_zRYw · 2023-08-21
> > > > > > > > > >
> > > > > > > > > > Dear Authors, Thank you for the reply. In the scenario where we have a few occurrences of a new event type, I wonder why it is not possible to very simply count its cooccurrences with other event types in quadratic time, and if it is, why it's impossible to compare with it as a baseline -- not saying you need to provide results now, just trying to understand the reasoning.
> > > > > > > > > >
> > > > > > > > > > My question about "why couldn't we detect time regularities by counting" was the first question in my original review. Your initial rebuttal suggested that "the NE vectors after the training stage are functionally equivalent to an approximate memory of all TR validity results after the counting process". Then counting should have naturally been a necessary baseline over which you need to demonstrate the claimed data efficiency and query efficiency, which are your claimed advantages that may or may not exist over counting methods. It's also natural to then ask for what benefits should people being willing to accept different degrees of approximation. If there was not enough time between the initial review and now, some initial comparison or discussion could be shown and more experiments could be proposed as future work. But in later conversations you suggested that NE didn't need to be compared with counting. You mentioned a few scenarios and claimed that counting was not natural but didn't explain why the straightforward counting was not applicable.
> > > > > > > > > >
> > > > > > > > > > Except for this question, I think this paper has multiple interesting aspects that other reviewers also pointed out. I acknowledge the novel aspects of your model, but have confusion about in what scenario NE will have wide use due to which advantage compared to counting methods. The novelty of the tasks and some technical details, on the other hand, are questioned by other reviewers. After re-evaluation, I find my initial question remains unsolved. I have lowered my confidence score since my doubts don't have enough time to be thoroughly addressed; I realize and sincerely apologize that I joined the conversation late during the permitted rebuttal period due to my untimely health issue and didn't expect that my initial question was something you acknowledge in the rebuttal but didn't provide a solution for. I believe it's for the paper's benefit to express my concerns, although others may have different evaluation.

---

> > > > > > > > > > > ### Author Response · Authors · 2023-08-22
> > > > > > > > > > >
> > > > > > > > > > > We sincerely appreciate the reviewer for the active discussion and insightful suggestions.
> > > > > > > > > > >
> > > > > > > > > > > We do not conduct experiments over the counting baselines because this paper is not motivated from application purposes. In this paper, we aim to advance the representation learning field, where our main contribution lies. Therefore, we prefer to leave room for future research. For example, future research may detailedly discuss NE's application advantage in certain resource constraint scenarios, as the reviewer insightfully suggested. Future research may also detailedly discuss NE's superiority in the accuracy-efficiency trade-off, as the reviewer kindly suggested.
> > > > > > > > > > >
> > > > > > > > > > > We appreciate the reviewer again for enabling a valuable rebuttal. We would like to highlight that contents in this paper are sufficient to support our claimed contributions.

---

> > > > > > > > > > > > ### Comment · Reviewer_zRYw · 2023-08-22
> > > > > > > > > > > >
> > > > > > > > > > > > Dear Authors, thanks for you reply. After spending some time revisiting your claimed contributions, I believe the main contribution of the paper is claimed to be data-efficient training and query efficiency, which indicate that the paper is absolutely application driven. Like most representation learning works, this paper aims to show that a structure with a bias is suitable for a task.
> > > > > > > > > > > > - The task in this paper, 1-1 TR detection and querying, is a tractable problem, an extension to existing tasks, and simpler than visual, language, or graph data processing.
> > > > > > > > > > > > - The presented method makes multiple simplifying assumptions that restrict the expressivity of the model and are not clearly presented as Reviewer 7yn1 mentioned. The choice of certain score and energy functions may cause unwanted biases but are not carefully investigated.
> > > > > > > > > > > > - While constraining the scopes in terms of the task and model mentioned above is okay, to show efficiency benefits one needs to at least present comprehensive comparison with obvious baselines. Simple counting appears tractable, to have the least unwanted biases the most accuracy possible, and fast in querying for 1-1 TR. The reason for using embeddings for TR detection and when embeddings are better for TR querying are not carefully illustrated.
> > > > > > > > > > > >
> > > > > > > > > > > > As a result, advances to the representation learning field feels like overclaiming due to the constrained and tractable task setting, insufficiently explained model assumptions and alternatives, and insufficiently justified advantages.

---

> ### Comment · Area_Chair_W4BB · 2023-08-18
>
> Hi Reviewer,
>
> This paper has divergent scores. So, please give your feedback after reading author rebuttal and other reviewers' comments.
>
> Your AC

---

> ### Author Response · Authors · 2023-08-21
> **Willingness to answer further questions**
>
> Dear reviewer zRYw
>
> We thank you for your precious time and constructive comments. As the discussion period will end soon, we are not sure whether our responses have addressed your questions. If you still have any questions about our work, we are more than happy to provide further responses for you.

---

### Official Review · Reviewer_Uoii · 2023-07-11

**Soundness:** 2 fair
**Presentation:** 1 poor
**Contribution:** 1 poor
**Rating:** 4
**Confidence:** 3

**Summary:**

This paper defined the complementary problems of TR detection and TR query, formulated their evaluation metrics, and adopted classic datasets for evaluations. Towards the TR problem, this paper proposed Noether Embedding (NE), which for the first time, enabled both the data-efficient formation and rapid retrieval of TRs simply through embedding each event sample.

**Strengths:**

1) This paper targeted at detecting and encoding temporal regularities (TRs) in events, which are of great importance.

2) This paper introduced Fourier basis to intrisincally model temporal regularities.

**Weaknesses:**

1) The authors overclaimed the first contribution.
The problems of TR detection and TR query are closely related to temporal assoication mining, with different evaluation metrics. However, these metrics were commonly adopted in other fields. So the problems were not new.

2) No strong baselines were compared in the experiments.
All compared baselines were not targeted this problem or had different assumptions, and therefore did not support the superiority of NE.

3) The paper is poorly presented.
The tasks were not presented clearly.
In addition this paper defined too many symbols but did not well explained them, which made readers lost.

4) The connection between Noether’s theorem and the proposed method is unclear and weak.
As an evidence, keywords such as "noether’s theorem" and "conservation law" only appeared in two paragraphs and I can't find how "noether’s theorem" motivated the formulation of Noether Embedding in a concrete description.


**Questions:**

1) The paper emphasized "distributed" representation was an advantage of NE. Is it a general property of knowledge graph embedding?

2) The paper also emphasized the proposed TR formulation was "data-efficient". Is this property brought by Fourier expansion? In opposite, what is the drawback or limitation?

3) How to perform the tasks with NE during inference? It seemed unclear in the paper.

**Limitations:**

1) The limitation of Fourier expansion for real-world temporal regularities was not discussed.

2) Potential negative societal impact was not discussed, e.g. causality of TR, privacy issue brought by TR detection.

---

> ### Author Rebuttal · Authors · 2023-08-09
>
> We sincerely appreciate the reviewer for spending valuable time reviewing our manuscript and providing insightful comments. We have improved our paper accordingly but also discovered some misunderstandings concerning the content and contributions of the work. Our responses are as below.
>
> **From ‘Weaknesses’**
>
> **Q1**: ‘The authors overclaimed the first contribution… the problems were not new’
>
> **A1**: **To our best knowledge, the problem is new. The detailed answer is provided in the first section of our global response at the top of the webpage.**
>
> **Q2**: ‘No strong baselines were compared in the experiments…therefore did not support the superiority of NE.’
>
> **A2**: This is because our problem is new, so there are no existing baselines, to our best knowledge, that exactly match our approach in the available research fields. Besides, NE's main superiority of 'efficiency' is evident even without comparative experiments. **The detailed answer is provided in the second section of our global response.**
>
> **Q3**: ‘The paper is poorly presented ... too many symbols but did not well explained them, which made readers lost.’
>
> **A3**: We have accordingly revised our paper. For example, we provide intuitive descriptions of the defined terms, such as explaining that standard confidence can be viewed as the probability that the head event will occur at time $t+\bigtriangleup$ once the body event occurs at time $t$.
>
> **Q4**: ‘I can't find how "noether’s theorem" motivated the formulation of Noether Embedding in a concrete description’
>
> **A4**: The motivation is described by the underlined text in Section 3.1. Specifically, Noether’s theorem inspires us with three structural biases when constructing NE: (1) the event embedding $\pmb{q}(t;ev)$ is constructed to make each local energy $g$ invariant to $t$; (2) the training loss is constructed to make the value of $g$ approximate TR validity; (3) the local energy $g$ is used as the decoding function. We have made the connection clearer in the revised version of the paper.
>
> **Q5**: ‘The connection between Noether’s theorem and the proposed method is unclear and weak’
>
> **A5**: The connection is evident in three aspects. Firstly, Noether’s theorem directly motivates the construction of NE, as shown in Section 3.1. Secondly, we attribute NE’s unique efficient learning capability directly to the Noether-inspired structural biases. Specifically, the first bias enables the data-efficient formation of TRs, the second bias mainly contributes to accurate TR detection and query, and the third bias directly leads to the rapid retrieval of TRs. Detailed explanations are in Section 3.3 and Appendix B.2. Thirdly, a more strict correspondence between NE variables and those in a physical system is shown in Appendix B.1.1.
>
> **From ‘Questions’**
>
> **Q1**: ‘The paper emphasized "distributed" representation was an advantage of NE. Is it a general property of knowledge graph embedding’
>
> **A1**: NE and knowledge graph embeddings utilize distinct aspects of distributed representations. Knowledge graph embeddings leverage their generalization (by interpolation) capabilities to achieve good performance in the completion task. NE, instead, use complex vectors to apply Fourier-like expansions, thus fitting and storing both event occurrences and TR validities in the embedding space. Specifically, the score function $f(t;ev)=\sum _{j=1}^d Real(\pmb{u} \circ e^{i \pmb{\omega} t}) _j$ and the decoding function $g(\tau;ev _b,ev _h)=2-2\sum _{j=1}^d Real(\overline{\pmb{u} _b} \circ \pmb{u} _h \circ e^{i \pmb{\omega} \tau}) _j\in [0,4]$ can be seen as Fourier-like expansions. The global time vector $\pmb{\omega}$ serves as the expansion basis, and event type vector $\pmb{u}$s store the coefficients for $f(t)$ (event occurrence) and composing $\overline{\pmb{u} _b} \circ \pmb{u} _h$ as the coefficients for $g(\tau)$ (TR validity).
>
> **Q2**: ‘The paper also emphasized the proposed TR formulation was "data-efficient". Is this property brought by Fourier expansion?’
>
> **A2**: We attribute NE’s data-efficient capability mostly to the fact that $g$ is invariant to $t$, which is directly inspired by Noether’s theorem. Detailed explanations are in Section 3.3. Fourier-like expansion, instead, is necessary for allowing good fitting and large-capacity storage, serving as a prerequisite for NE to perform well on large-scale real-world datasets used in the paper.
>
> **Q3**: ‘How to perform the tasks with NE during inference’
>
> **A3**: As described in Section 3.2, $\mathop{\min}\limits _{\tau\in\mathbb{T} _r} g(\tau)$ is computed, which is compared with a global threshold $g _{th}$ to decide whether a potential TR is valid or not (for TR detection). For a valid TR, the $\tau'$ which minimizes $g(\tau), \tau \in \mathbb{T} _r$ is selected as the model output of the relative time (for TR query). We have provided clarifications on the relationship between NE decoding and TR detection and TR query in the revised paper.
>
> **From ‘Limitations’**
>
> **Q1**: ‘The limitation of Fourier expansion for real-world temporal regularities was not discussed’
>
> **A1**: In Appendix B.2.2, we have theoretically analyzed the requirement for the vector dimension $d$ to be larger than the number of absolute time points $T _a$ to avoid significant performance degradation of NE, as observed in the GDELT dataset. This limitation imposes a storage capacity constraint for large datasets. We have included notifications regarding this limitation in the revised paper.
>
> **Q2**: ‘Potential negative societal impact was not discussed, e.g. causality of TR, privacy issue brought by TR detection.’
>
> **A2**: Thanks for your reminding. We have accordingly revised our paper to address these concerns.
>
> We believe that our work makes a nontrivial contribution to the representation learning community by developing NE as a first efficient TR learner with event embeddings and proposing tasks to fairly evaluate embeddings' TR learning capabilities.

---

> ### Comment · Area_Chair_W4BB · 2023-08-18
>
> Hi Reviewer,
>
> This paper has divergent scores. So, please give your feedback after reading author rebuttal and other reviewers' comments.
>
> Your AC

---

> ### Author Response · Authors · 2023-08-21
> **Willingness to answer further questions**
>
> Dear reviewer Uoii
>
> We thank you for your precious time and constructive comments. As the discussion period will end soon, we are not sure whether our responses have addressed your questions. If you still have any questions about our work, we are more than happy to provide further responses for you.

---

### Official Review · Reviewer_tTxw · 2023-07-18

**Soundness:** 2 fair
**Presentation:** 3 good
**Contribution:** 2 fair
**Rating:** 3
**Confidence:** 3

**Summary:**

This paper introduced a new task, $\textit{temporal regularity mining}$,  and proposed a Noether Embedding  to rapidly retrieval TR.

**Strengths:**

- The argued temporal regularities sound interesting.
- Good writing.

**Weaknesses:**

- The proposed temporal regularity mining is not a new task with a new paradigm, which the existing methods cannot well tackle.  I think it is more like an extended task based on event embedding using temporal knowledge graph data.
- All existing event embedding methods can be used to tackle the proposed TR mining task. However, there is no discussion to clarify why these existing methods cannot well deal with this new task.
- Mining general TR sounds interesting and useful, but the mined TR is not general. In other words, it is more like a temporal association between events. For example, as shown in the supplementary material, (China, Appeal for diplomatic cooperation, Malaysia) $\rightarrow$ (South Korea, Express intent to settle dispute, China). An ideal TR should be no matter who does the body event, and then the head body will occur soon, which is invariant to the time.
- The temporal range between the body and head events is large, from several days to several years. So how to set the $\tau, \eta$?  Please discuss this detail.
- More experiments on traditional tasks in event embedding, such as event prediction, should be conducted to further demonstrate the effectiveness of the proposed method.

**Questions:**

Please see the Weaknesses.

**Limitations:**

Yes

---

> ### Author Rebuttal · Authors · 2023-08-09
>
> We sincerely appreciate the reviewer for spending valuable time reviewing our manuscript. However, there do exist many factual errors, which are justified below.
>
> **From ‘Weaknesses’**
>
> **Q1**: ‘The proposed temporal regularity mining is not a new task with a new paradigm’
>
> **A1**: To our best knowledge, the problem is new. Our main problem is how to enable event embeddings with an efficient TR learning capability, which is distinct from those in the temporal rule mining field. Specifically, temporal rule mining is typically studied for practical applications, aiming to uncover event regularities in specific domains [1] [2] [3]. Our goal, instead, is to advance the representation learning field by enabling embedding models to efficiently learn the atomic structure (TR) of event schemas, similar to how humans learn [4] [5]. We, therefore, develop NE with such a novel capability and propose two complementary tasks for evaluating embedding capabilities. Our TR tasks are defined with evaluation metrics borrowed from the rule mining field [6] only to guarantee fair and reasonable evaluations.
>
> **Q2**: ‘I think it is more like an extended task based on event embedding using temporal knowledge graph data.’
>
> **A2**: We would like to clarify that we use temporal knowledge graph data because they are classic and authoritative, allowing for fair evaluations. However, our problem is targeted at the most basic form of events $(ev, t)$, which serves as the foundation for our definition of TR.  It is important to note that both of our proposed tasks and method can easily generalize to arbitrary forms of structured events, and temporal knowledge graph is just a special case. For example, by setting each (s,p,o) as an event type, our tasks and NE method can handle temporal knowledge graph data in the form of (s,p,o,t). Similarly, by setting (s,p) as an event type, our tasks and NE method can handle data in the form of (s,p,t), and so on.
>
> **Q3**: ‘All existing event embedding methods can be used to tackle the proposed TR mining task’
>
> **A3**: Existing embeddings, indeed, cannot tackle the proposed tasks in two levels. Firstly, the ‘efficient’ learning capability is unique to NE and not present in existing embeddings. We attribute this unique capability of NE to its structural biases inspired by Noether’s theorem. Specifically, (1) the event embedding $\pmb{q}(t;ev)$ is constructed to make local energies $g$s invariant to $t$; (2) the training loss is constructed to make the value of $g$ approximate TR validity; (3) the local energy $g$ is used as the decoding function. Secondly, even when setting aside the ‘efficient’ requirement, existing embeddings still cannot learn TRs accurately, as demonstrated by experiments in Table 1 and evaluated by the proposed TR detection and query tasks. This is primarily because they over-apply the generalization capabilities of distributed representations, which hinders the fit of event occurrences, as discussed in Section 4.2.
>
> **Q4**: ‘the mined TR is not general’
>
> **A4**: We would like to clarify that NE can indeed learn ‘general’ TRs when we set (s,p,o,t) with the same p (predicate) to denote an event type, rather than with the same (s,p,o) as set in the paper. This further proves the wide range of potential applications for NE, and we appreciate the reviewer for suggesting this. We have emphasized this potential in the revised paper.
>
> **Q5**: ‘how to set the $\tau, \eta$? Please discuss this detail’
>
> **A5**: We have provided detailed explanations in Section 3.2 and Section 4.1 on how to set the values of $\tau$ and $\eta$. In our experiments, $\tau$ is traversed through set $\mathbb{T} _r$ of the relative time points such as $\mathbb{T} _r: \{-\tau _{max},...,0, ..., \tau _{max}\}$ to plot the decoding results. We set $\tau _{max}=T _a-1$. As for $\eta$, we set it to 0.1 in $\bigtriangleup$s for strict evaluations and take the upper integer $\bigtriangleup=[\tau-\lceil \tau\eta \rceil, \tau+\lceil\tau\eta\rceil]$. It is important to note that even in extreme situations where both body and head event occurrences are equal to 2, stochastic noises are still unlikely to interfere with the evaluation of TR validity since $\eta=0.1$ is strict.
>
> **Q6**: ‘More experiments on traditional tasks in event embedding, such as event prediction, should be conducted to further demonstrate the effectiveness of the proposed method’
>
> **A6**: We appreciate the reviewer for making such a suggestion. However, we have already shown that NE is the first efficient TR learner with event embeddings. Specifically, only NE can encode TRs from limited event items (as shown in Figure 4) and rapidly retrieve TRs (by applying $g(\tau)$). Such a uniqueness is fundamental and qualitative, requiring no comparative experiments to demonstrate. Our comparative experiments have further demonstrated NE's superiority over existing embeddings when setting aside the 'efficient' requirement to only learn TRs accurately. Therefore, we believe that our experiments are sufficient to support the effectiveness of NE.
>
> Thank you for your valuable feedback, and we have made sure to address the issues raised in your review.
>
> **References**
>
> [1] Segura‐Delgado A, Gacto M J, Alcalá R, et al. Temporal association rule mining: An overview considering the time variable as an integral or implied component. DMKD, 2020.
>
> [2] Chen M, Chen S C, Shyu M L. Hierarchical temporal association mining for video event detection in video databases. IEEE, 2007.
>
> [3] Yoo J S, Shekhar S. Similarity-profiled temporal association mining. IEEE, 2008.
>
> [4] Pudhiyidath A, Roome H E, Coughlin C, et al. Developmental differences in temporal schema acquisition impact reasoning decisions. Cognitive Neuropsychology, 2020.
>
> [5] Chaudhuri R, Fiete I. Computational principles of memory. Nature neuroscience, 2016.
>
> [6] Galárraga L, Teflioudi C, Hose K, et al. Fast rule mining in ontological knowledge bases with AMIE+. The VLDB Journal, 2015.

---

> > ### Comment · Area_Chair_W4BB · 2023-08-17
> >
> > Dear Reviewers,
> >
> > Please have a careful look at the author response and give a feedback, as your score is lower than others.
> >
> > Your AC

---

> ### Author Response · Authors · 2023-08-21
> **Willingness to answer further questions**
>
> Dear reviewer tTxw
>
> We thank you for your precious time and constructive comments. As the discussion period will end soon, we are not sure whether our responses have addressed your questions. If you still have any questions about our work, we are more than happy to provide further responses for you.

---

### Official Review · Reviewer_w2A9 · 2023-07-24

**Soundness:** 3 good
**Presentation:** 3 good
**Contribution:** 3 good
**Rating:** 6
**Confidence:** 3

**Summary:**

This paper introduces Noether Embedding (NE), a new model for efficient learning of temporal regularities with event embeddings. Experiments conducted on three datasets show the superior performance of this work compared to classic embeddings in detecting valid TRs and querying TR intervals.


**Strengths:**

1. The authors introduce Noether Embedding (NE), a new model that enables the data-efficient formation and rapid retrieval of temporal regularities simply through embedding each event sample. NE possesses the intrinsic time-translation symmetries of TRs, which facilitates TR encoding insensitive to sample size and TR retrieval in constant time complexity. This is a novel approach that has not been explored in previous works.
2. The authors formally define complementary problems of TR detection and TR query, formulate their evaluation metrics, and evaluate NE on classic ICEWS14, ICEWS18, and GDELT datasets. This is a rigorous evaluation of the proposed model and provides evidence of its superior performance compared to classic embeddings with additional calculation efforts.
3. The paper is well-written and clear, with a concise abstract and introduction that provide a good overview of the problem and the proposed solution. The authors also provide detailed explanations of the model and the evaluation metrics.


**Weaknesses:**

1. In Table 2 of the Appendix, the recall rate of NE is lower than that of TASTER. It would be better if the reason could be explained.
2. It should be further explained why gc(τ) in line 79 has different input from gc(tr) in formula 3.

**Questions:**

1. Table 2 in the Appendix shows that NE has extremely high accuracy, while the recall rate is not the highest. May I know the reason for this phenomenon?
2. Could you further explain why NE has an overwhelming advantage over all baselines?


**Limitations:**

The author proposes the limitations of NE and solutions in terms of storage efficiency in lines 307-311.
1. In the future, methods will be explored to store event occurrences and time patterns in different regions to improve the storage efficiency of NE.
2. Future research will explore methods to compose 1-1 TRs into graphical and hierarchical event schemas and combine NE with deep learning and reinforcement learning methods.

---

> ### Author Rebuttal · Authors · 2023-08-09
>
> We sincerely appreciate the reviewer for spending valuable time reviewing our manuscript and providing insightful comments. We have improved our paper accordingly and our responses are as below.
>
> **From ‘Weaknesses’**
>
> **Q1**: ‘In Table 2 of the Appendix, the recall rate of NE is lower than that of TASTER. It would be better if the reason could be explained.’
>
> **A1**: The reason is that we report the highest F1 score of each model in comparative studies by tuning their respective global threshold, denoted as $g _{th}$. As the F1 score is calculated using the formula $F1 = \frac{2 * precision * recall}{precision + recall}$, TASTER achieves its highest F1 score by reporting many false positives, resulting in a relatively high recall rate but an extremely low precision rate. We have included this explanation in the revised Appendix and emphasized the ‘highest F1 score’ evaluation in the revised paper.
>
> **Q2**: ‘It should be further explained why gc(τ) in line 79 has different input from gc(tr) in formula 3.’
>
> **A2**: We would like to clarify that they refer to the same gc but with different emphases. On the one hand, the $gc(tr)$ in Formula 3 represents the definition of global confidence for each tr. On the other hand, since $tr: (ev _b, ev _h, \bigtriangleup) = (ev _b, ev _h, \tau, \eta)$, we have $gc(tr)=gc(ev _b, ev _h, \tau, \eta)= gc(\tau; ev _b, ev _h, \eta)$. Therefore, the $gc(\tau)$ in line 79 emphasizes the fact that $gc$ can be expressed as a function of $\tau$ with fixed $ev _b, ev _h, \eta$. We have clarified this further in the revised paper.
>
> **From ‘Questions’**
>
> **Q1**: ‘Table 2 in the Appendix shows that NE has extremely high accuracy, while the recall rate is not the highest. May I know the reason for this phenomenon?’
>
> **A1**: The reason is that we report the highest F1 score of each model in comparative studies by tuning their respective global threshold, denoted as $g _{th}$. As the F1 score is calculated using the formula $F1 = \frac{2 * precision * recall}{precision + recall}$, NE achieves its highest F1 score by reporting few false positives, resulting in a relatively high precision rate but a relatively low recall rate. We have emphasized the ‘highest F1 score’ evaluation in the revised paper.
>
> **Q2**: ‘Could you further explain why NE has an overwhelming advantage over all baselines?’
>
> **A2**: Firstly, the 'efficient' learning capability is unique to NE, which is fundamental and qualitative, requiring no comparative experiments to demonstrate. Specifically, only NE can encode TRs from limited event items (as shown in Figure 4) and rapidly retrieve TRs (by applying $g$). We attribute this mainly to three structural biases inspired by Noether's theorem: (1) the event embedding $\pmb{q}(t;ev)$ should be constructed to make each local energy $g$ invariant to $t$; (2) the training loss should be constructed to make the value of $g$ approximate TR validity; (3) we should use local energy $g$ as the decoding function. Without such structural biases, baseline embeddings can not learn TRs efficiently.
>
> When setting aside the 'efficient' requirement by added with the same interface $g'$, baseline models still can not learn TRs accurately as compared to NE, as demonstrated by the comparative experiments. The main reason is that baseline models over-apply the generalization capabilities of distributed representations, which hinders the fit of event occurrences, as discussed in Section 4.2 with Figure 3.
>
> Once again, we would like to express our gratitude to the reviewer for the valuable feedback, which has helped us further improve our manuscript.

---

> ### Author Response · Authors · 2023-08-21
> **Willingness to answer further questions**
>
> Dear reviewer w2A9
>
> We thank you for your precious time and constructive comments. As the discussion period will end soon, we are not sure whether our responses have addressed your questions. If you still have any questions about our work, we are more than happy to provide further responses for you.

---

### Official Review · Reviewer_bn9B · 2023-07-25

**Soundness:** 3 good
**Presentation:** 3 good
**Contribution:** 3 good
**Rating:** 6
**Confidence:** 4

**Summary:**

In this paper, the authors introduce the concept of temporal regularities (TR), which indicates temporal associations invariant to time shifts between events. The authors claim that existing models are lack of the TR learning capability. Based on this idea, the authors define two tasks, TR detection and TR query, as well as their evaluation metrics. They also further develop a new framework to learn a set of event representations regularized by fixed time embeddings. Experiments on several benchmark datasets demonstrate the effectiveness of the proposed framework and its superiority on TR learning compared to existing methods.

**Strengths:**

* The proposed temporal regularity problem is important and many existing models are lack of such TR learning capabilities.
* The proposed TR detection and TR query tasks are well-designed. The corresponding evaluation metrics are also reasonable.
* The proposed solution is simple and effective. Experiments demonstrate significant improvement on TR compared to previous methods.

**Weaknesses:**

* The event embedding implementation part is not very clear. Details are insufficient for reimplementation.
* Font sizes in Figure 5 and 6 are too tiny.

**Questions:**

* In Line 166-168, each (s,p,o) combination corresponds to a specific event embedding. From the authors' code, it seems like the authors have tried different embedding strategy, such as encoding s, p and o separately, or encoding (s,p,o) as a whole. What's your choice among these different implementations?
* In formula 7, the conserved local energy is only relevant two events. If the relative time is conditioned on other events, could this framework handle it?
* I'm interested how much have the model learnt from statistic prior. Have you tried to compute the averaged relative time for every event pair in the training set and then use such averaged relative time for testing?


**Limitations:**

The authors have adequately address the limitations.

---

> ### Author Rebuttal · Authors · 2023-08-09
>
> We sincerely appreciate the reviewer for spending valuable time reviewing our manuscript and providing insightful comments. We have improved our paper accordingly and our responses are as below.
>
> **From ‘Weaknesses’**
>
> **Q1**: ‘The event embedding implementation part is not very clear. Details are insufficient for reimplementation.’
>
> **A1**: Original details of model implementation include: the determination of hyperparameters (in Section 4.1) and the training details (in Appendix C.1). We have added justifications of how NE decoding relates to TR detection and TR query (in Section 3.2) in the revised paper. Specifically, $\mathop{\min}\limits _{\tau\in\mathbb{T} _r} g(\tau)$ is computed, which is compared with a global threshold $g _{th}$ to decide whether a potential TR is valid or not (for TR detection). For a valid TR, the $\tau'$ which minimizes $g(\tau), \tau \in \mathbb{T} _r$ is selected as the model output of the relative time (for TR query).
>
> **Q2**: ‘Font sizes in Figure 5 and 6 are too tiny.’
>
> **A2**: We have taken note of this feedback and have made the necessary changes by enlarging the font sizes in Figure 5 and 6 in the revised paper.
>
> **From ‘Questions’**
>
> **Q1**: ‘From the authors' code, it seems like the authors have tried different embedding strategy, such as encoding s, p and o separately, or encoding (s,p,o) as a whole. What's your choice among these different implementations?’
>
> **A1**: We appreciate the reviewer's careful observation. Our choice of the current form of NE is based on its ability to fit large datasets effectively while still maintaining the efficient learning capability. We have found that encoding s, p, and o separately leads to a higher loss after training convergence. This reduces the capacity of NE for large datasets and subsequently affects its performance in both TR detection and TR query. Making the model smaller while maintaining effectiveness is an area that can be explored in future work.
>
> **Q2**: ‘If the relative time is conditioned on other events, could this framework handle it?’
>
> **A2**: One advantage of NE is exactly its ability to efficiently store large amounts of interlinked TRs. We attribute such a large capacity of NE storage to its Fourier-like representations. Specifically, the score function $f(t;ev)=\sum _{j=1}^d Real(\pmb{u} \circ e^{i \pmb{\omega} t}) _j$ and the decoding function $g(\tau;ev _b,ev _h)=2-2\sum _{j=1}^d Real(\overline{\pmb{u} _b} \circ \pmb{u} _h \circ e^{i \pmb{\omega} \tau}) _j\in [0,4]$ can be viewed as Fourier-like expansions. The global time vector $\pmb{\omega}$ provides the expansion basis, while the event type vectors $\pmb{u}$s store the coefficients for $f(t)$ (revealing event occurrence) and compose $\overline{\pmb{u} _b} \circ \pmb{u} _h$ as the coefficients for $g(\tau)$ (revealing TR validity).
>
> **Q3**: ‘I'm interested how much have the model learnt from statistic prior. Have you tried to compute the averaged relative time for every event pair in the training set and then use such averaged relative time for testing?’
>
> **A3**: While we have not specifically explored this setting, we have demonstrated the flexibility of NE through a grouped experiment. We have grouped valid TRs based on their golden relative time and showcased NE's performance in TR query. The results in Figure 4 (c) indicate that NE performs consistently well in learning TRs with varying $\tau$s. We have not tested TR detection using grouped $\tau$s as the golden relative time for invalid TRs does not hold meaning.
>
> Once again, we would like to express our gratitude to the reviewer for the valuable feedback, which has helped us further improve our manuscript.

---

> ### Author Response · Authors · 2023-08-21
> **Willingness to answer further questions**
>
> Dear Reviewer bn9B
>
> We thank you for your precious time and constructive comments. As the discussion period will end soon, we are not sure whether our responses have addressed your questions. If you still have any questions about our work, we are more than happy to provide further responses for you.

---

### Author Rebuttal · Authors · 2023-08-09

Three main justifications are provided below.

## 1. Novelty of the problem

**Q**: ‘The authors overclaimed the first contribution…the problems were not new’

**A**: To our best knowledge, the problem is new. Our main problem is how to enable event embeddings with an efficient TR learning capability, which is distinct from those in the temporal rule mining field. Specifically, temporal rule mining is typically studied for practical applications, aiming to uncover event regularities in specific domains [1] [2] [3]. Our goal, instead, is to advance the representation learning field by enabling embedding models to efficiently learn the atomic structure (TR) of event schemas, similar to how humans learn [4][5]. Our problem is not only new but also important because (1) embedding models are a promising technology field; (2) TRs are a basic world structure among events; (3) efficient learning is a long-pursued human-like capability.

To address this problem, we have developed NE as the first efficient TR learner using event embeddings. Additionally, we have proposed two complementary tasks for evaluating the TR learning capabilities of embedding models. Our tasks are defined with evaluation metrics borrowed from the rule mining field [6], only to ensure fair and reasonable evaluations.

To avoid misunderstandings, we have reversed the order of our two contributions, making the development of NE the first contribution and the proposal of tasks the second. Both of these contributions aim to advance the representation learning field.

## 2. Fairness of comparisons

**Q**: ‘The paper would be greatly strengthened by a clear justification of why this is the fairest available comparison between methods.’

**A**: Our main contribution is developing NE as a first ‘efficient’ TR learner with event embeddings. The uniqueness of NE in terms of efficiency is fundamental and qualitative, requiring no comparative experiments to demonstrate. Specifically, only NE can encode TRs from limited event items (as shown in Figure 4) and rapidly retrieve TRs (by applying $g(\tau)$), thanks to its specific structural biases. Existing embedding models lack these capabilities.

Only when setting aside the ‘efficient’ requirement are comparative experiments necessary to compare the TR learning accuracy between NE and existing embeddings. This is secondary compared to the ‘efficient’ distinction. Since there are, to our best knowledge, no existing embedding baselines that exactly match our requirements in the available research fields, we have made efforts to ensure fairness in comparison through various means:

(1)	Models are applied in the same way. We input the same event data during the training stage and add the same interface $g’(\tau)$ to the respective model outputs of score functions. This interface is applied to both NE and the embedding baselines, in the same manner, to indirectly compute TR validity from stored event occurrences. The excellent performance of NE with this interface validates its effectiveness.

(2)	Evaluations are reliable. The evaluation metrics used in our proposed tasks are borrowed and adapted from the mature field of rule mining [6]. This guarantees a reliable evaluation of the TR learning capabilities of embedding models.

(3)	Baselines are classic.  We have chosen baselines from the well-developed field of temporal knowledge graph embedding, which has a wide range of classic embeddings for structured events.

(4)	Dataset is convincing. Our main experiments are conducted on three classic real-world event datasets.

(5)	Performance is explainable. We provide detailed explanations in the paper regarding why NE works and why existing embeddings do not, both theoretically and experimentally.

We have further clarified these points in the revised paper.

## 3. Why NE works and overwhelms

The success of NE can be attributed to two main factors: the Noether-inspired structural biases and the Fourier-like memory representations. One contributes to NE’s efficient TR learning capability while the other enables NE’s large-capacity storage for both TR validity and event occurrences.

(1)	The Noether-inspired structural biases

The Noether-inspired structural biases can be summarized as below: (i) the event embedding $\pmb{q}(t;ev)$ is constructed to make each local energy $g$ remain invariant to $t$; (ii) the training loss is designed to make the value of $g$ approximate TR validity; (iii) the local energy $g$ is used as the decoding function.

(2)	The Fourier-like memory representations

The score function $f(t;ev)=\sum _{j=1}^d Real(\pmb{u} \circ e^{i \pmb{\omega} t}) _j$ and the decoding function $g(\tau;ev _b,ev _h)=2-2\sum _{j=1}^d Real(\overline{\pmb{u} _b} \circ \pmb{u} _h \circ e^{i \pmb{\omega} \tau}) _j\in [0,4]$ can be viewed as Fourier-like expansions. The global time vector $\pmb{\omega}$ provides the expansion basis, while the event type vectors $\pmb{u}$s store the coefficients for $f(t)$ (revealing event occurrence) and compose $\overline{\pmb{u} _b} \circ \pmb{u} _h$ as the coefficients for $g(\tau)$ (revealing TR validity).

Detailed explanations can be found in Section 3.3 and Appendix B.2.

## Reference

[1] Segura‐Delgado A, Gacto M J, Alcalá R, et al. Temporal association rule mining: An overview considering the time variable as an integral or implied component. DMTD, 2020.

[2] Chen M, Chen S C, Shyu M L. Hierarchical temporal association mining for video event detection in video databases. IEEE, 2007.

[3] Yoo J S, Shekhar S. Similarity-profiled temporal association mining. IEEE, 2008.

[4] Pudhiyidath A, Roome H E, Coughlin C, et al. Developmental differences in temporal schema acquisition impact reasoning decisions. Cognitive Neuropsychology, 2020.

[5] Chaudhuri R, Fiete I. Computational principles of memory. Nature neuroscience, 2016.

[6] Galárraga L, Teflioudi C, Hose K, et al. Fast rule mining in ontological knowledge bases with AMIE+. VLDB, 2015.

---

> ### Author Response · Authors · 2023-08-20
> **Facts worth emphasizing**
>
> These two facts can **further assist reviewers** in evaluating our contributions effectively.
>
> 1. The use of NE vectors as a **'query-efficient memory'** for learned TRs is a key aspect of our approach. We store the validity of each potential TR: $(ev_b,ev_h,\tau)$ separately in each event-type vector $\pmb{u}(ev)$ during the training stage. By utilizing GPUs, we can retrieve this information in constant time ($O(1)$) by calculating $g=||\pmb{u}(ev_b)-\pmb{u}(ev_h) \circ \pmb{r}(\tau)||$. This distinguishes our problem from rule mining, as we **also focus on the storage of learned TRs**.
>
> 2. The data-efficient formation of structures has **traditionally been a challenge for distributed representations**. Deep learning methods typically require a large amount of lower-level data to learn higher-level structures, such as concepts learned in CNN from pictures [1] and grammars learned in GPT from sentences [2]. In our case, we tackle the learning of higher-level TRs in event embeddings from limited lower-level events (shown in Figure 4). Consequently, the problem we address is **not only important but also quite challenging.** We achieve this by leveraging embeddings with specifically-designed Noether-inspired structural biases.
>
> References
>
> [1] Chen, Zhi, Yijie Bei, and Cynthia Rudin. "Concept whitening for interpretable image recognition." Nature Machine Intelligence, 2020.
>
> [2] Mahowald, Kyle, et al. "Dissociating language and thought in large language models: a cognitive perspective." arXiv, 2023.

---

> > ### Author Response · Authors · 2023-08-21
> > **Major improvements (in the revised paper)**
> >
> > 1. **General Idea**: We have made significant improvements to our manuscript by **consistently highlighting the unique efficiency of NE throughout the paper**, emphasizing its relevance to the topic, methodology, and experimental results.
> >
> > 2. **Topic**: In order to **differentiate our research from rule mining**, we have modified our topic to ‘Noether Embedding: a **data-efficient learner and query-efficient memory** of temporal regularities’. This change better reflects the focus of our work.
> >
> > 3. **Introduction**: To provide a more comprehensive understanding of our problem, we have **added a discussion on the challenges** associated with data-efficient learning of structures in distributed representations. Additionally, we have **reorganized the order of our contributions**, now presenting the development of NE as the first contribution and the proposal of tasks as the second, to avoid any potential misunderstandings.
> >
> > 4. **Task**: We have placed more emphasis on the **motivation behind our proposed tasks**, highlighting their relevance in evaluating the temporal regularity learning capability of embedding models. Furthermore, we have provided **intuitive explanations for the defined terms**, such as clarifying that standard confidence represents the probability of the head event occurring at time $t+\bigtriangleup$ given the occurrence of the body event at time $t$.
> >
> > 5. **Method**: We have given greater emphasis to the three **Noether-inspired structural biases and their relationship to NE's unique properties**. Specifically, we have clarified that the data-efficient property is primarily attributed to NE’s innate time-translation symmetries, while the query-efficient property is achieved through the decoding of conserved local energies.
> >
> > 6. **Experiment**: We have restructured our experiments to **first demonstrate the data-efficient learning and query-efficient memory capabilities of NE**. Only after this demonstration, we then proceed to conduct comparative experiments that evaluate the temporal regularity learning accuracy of embedding models, **setting aside the efficiency requirement**.
> >
> > 7. **Conclusion**: As suggested by the reviewers, we have expanded the discussion on the **limitations and social impacts** of our approach in the conclusion section. This addition provides a more comprehensive understanding of the implications of our research.
> >
> > It is important to highlight that the **enhancements mentioned above are diectly derived from the contributions made to our initial version of the manuscript**, aimed at more effectively presenting the focus of our research.

---

> ### Author Response · Authors · 2023-08-22
> **One Significant Improvement**
>
> Inspired by a comment, **we have calculated the compression ratio of NE compared to exact counting**, based on the conducted experiments on ICEWS14 and ICEWS18 datasets in the paper. **In addition to the data-efficient and query-efficient properties, our new findings indicate that NE is, in fact, also a storage-efficient memory**. Here is a detailed analysis:
>
> The storage of NE vectors, denoted as $S(NE)$, can be calculated as follows: $S(NE) = S(ev-vector) + S(time-vector) = 2 * N * d * 64bit + 2 * d * 64bit$. In our experiments, we used torch.LongTensor and N represents the number of event-type vectors.
>
> On the other hand, the storage of exact counting, denoted as $S(Count)$, can be calculated as follows: $S(Count) = S(TR) + S(event) = N^2 * T_a * log_2 (n/N) bit + N * (n/N) * log_2 (T_a) bit$. Here, $n$ represents the number of all event occurrences. We reserved the storage accuracy of TR validity to effectively distinguish different values, resulting in approximately $log_2 (n/N) bit$ for each TR validity $(ev_b, ev_h, \tau)$.
>
> For the ICEWS14 and ICEWS18 datasets, where $d = 400, T_a = 365$, and $n = 90730,468558, N = 50295,266631$, **we calculated the compression ratio of NE as 421 and 2336, respectively**. This remarkable capability of NE can be attributed to the fact that it separately stores the information of TR validities $(ev_b, ev_h, \tau)$ using event-type vectors and a global time vector. By representing the common information of related TRs efficiently in memory, NE achieves a compression ratio that is **approximately linear to the number of event types $N$**. Our results demonstrate that **NE strikes a balance between storage efficiency and accuracy compared to exact counting**. Therefore, NE holds **significant potential for applications in storage-constrained scenarios**, as the comment has suggested. We have added this discussion in the revised paper as an extension.

---

### Decision · Program_Chairs · 2023-09-21

**Decision:**

Accept (poster)

**Comment:**

This paper receives mixed reviews. The reviewers express concern on the novelty of the problem, fairness of comparisons, unclear on the important aspects of the proposed method. The authors provide very detailed responses these reviewers comments. After reading the author rebuttal, two reviewers are very active in the discussion with authors and give very detailed and specific comments to authors. The authors also give very detailed explanation and new results on the compression ratios of NE compared to exact counting, which is appreciated by the reviewers. Another active reviewers are also persuaded by the main strengths of the paper and think this paper can be accepted. The AC carefully check the paper, the reviewer comments and the author responses, and downgrade the score of some less active reviewers. Overall, the AC agrees with the contribution of this paper and makes accept recommendation to this paper. The authors are required to add the new results in the rebuttal and discussion with reviewers to the final version.